# Interactomic Analyses and a Reverse Engineering Study Identify Specific Functional Activities of One-to-One Interactions of the S1 Subunit of the SARS-CoV-2 Spike Protein with the Human Proteome

**DOI:** 10.3390/biom14121549

**Published:** 2024-12-03

**Authors:** Giovanni Colonna

**Affiliations:** Unit of Medical Informatics—AOU Luigi Vanvitelli, University of Campania, 80138 Naples, Italy; giovanni.colonna@unicampania.it

**Keywords:** SARS-CoV-2, S1 subunit of the SARS-CoV-2 Spike protein, SARS-CoV-2 and cancer, one-to-one interactions in COVID-19 infection, TP53 and RSP27A, long COVID-19, COVID-19 and cancer

## Abstract

The S1 subunit of SARS-CoV-2 Spike is crucial for ACE2 recognition and viral entry into human cells. It has been found in the blood of COVID-19 patients and vaccinated individuals. Using BioGRID, I identified 146 significant human proteins that interact with S1. I then created an interactome model that made it easier to study functional activities. Through a reverse engineering approach, 27 specific one-to-one interactions of S1 with the human proteome were selected. S1 interacts in this manner independently from the biological context in which it operates, be it infection or vaccination. Instead, when it works together with viral proteins, they carry out multiple attacks on single human proteins, showing a different functional engagement. The functional implications and tropism of the virus for human organs/tissues were studied using Cytoscape. The nervous system, liver, blood, and lungs are among the most affected. As a single protein, S1 operates in a complex metabolic landscape which includes 2557 Biological Processes (GO), much more than the 1430 terms controlled when operating in a group. A Data Merging approach shows that the total proteins involved by S1 in the cell are over 60,000 with an average involvement per single biological process of 26.19. However, many human proteins become entangled in over 100 different biological activities each. Clustering analysis showed significant activations of many molecular mechanisms, like those related to hepatitis B infections. This suggests a potential involvement in carcinogenesis, based on a viral strategy that uses the ubiquitin system to impair the tumor suppressor and antiviral functions of TP53, as well as the role of RPS27A in protein turnover and cellular stress responses.

## 1. Introduction

Understanding COVID-19 is an ongoing challenge because of its complex pathology and limited molecular insights [1,2]. Many clinicians rely on treating symptoms similarly to other viral diseases [3], which complicates a comprehensive understanding of the clinical picture, especially regarding “long COVID” (PCS) [4], where patients experience persistent symptoms for months [5]. Despite four years of research, defining the characteristics of long COVID-19 remains difficult. PCS exhibits diverse symptoms like “brain fog”, immune dysregulation, and fatigue, which affect quality of life [5]. The lack of understanding of the molecular mechanisms behind PCS limits treatments to symptom management rather than targeted therapies [6]. Controlled clinical trials often lack in-depth metabolic research [6].

One of the key biological characteristics of SARS-CoV-2, as well as several other viruses, is spike proteins that allow these viruses to penetrate host cells and cause infection. The SARS-CoV-2 Spike is a glycoprotein, which forms a trimer anchored to the membrane by its transmembrane segment. Upon ACE2 binding, the pre-fusion Spike undergoes conformational rearrangements that promote the S1:ACE2 complex dissociation from S2 to drive membrane fusion, assuming a stable post-fusion conformation. Therefore, the S1 subunit plays a critical role in binding to ACE2 and has various other functions [7]. It is often referred to as a “spikeopathy” [8]. Its immunogenic characteristics remain the same even when modified during vaccine production [9]. However, its interactions with human proteins during infection are less understood. While with the vaccine it acts alone by interacting with human proteins, in the infection it attacks human proteins also together with other viral proteins [10,11]. However, there are also cases where it attacks specific human proteins with a one-to-one interaction [10]. In this study, we use the term ‘one-to-one interaction’ to describe the specific binding of the S1 subunit to target proteins (more details in Appendix A). However, protein–protein interaction (PPI) studies like those in BioGRID provide insights into S1 interactions, but it is essential to validate these in complex metabolic contexts since factors like the cellular environment and post-translational modifications influence these interactions.

The widespread presence of various S1 proteoforms complicates distinguishing responses from vaccination and infection. S1 has many sites for post-translational modifications (PTMs), such as K, T, S, and Y, scattered throughout the structure [12] that complicate understanding. Although mRNA vaccines have been lifesaving, they can cause adverse effects linked to the S protein’s interaction with human tissues. Studies have detected significant levels of circulating S1 protein after both vaccination and infection [13,14,15,16,17]. However, linking these S1 levels to observed symptoms is difficult because of non-linear biological responses.

Understanding the metabolic organization of COVID-19 requires recognizing the constraints and rules governing biological networks, where proteins play a crucial role through the metabolic interactions. Graph theory aids in representing these networks and their biological functions, clarifying the complexities of disease progression. I believe that understanding these aspects is possible by using relevant protein biosignatures as seeds for constructing PPINs (Protein–Protein Interaction Networks). The interactions are the crucial step in this analysis. They cannot be estimated indirectly but must be detected using the experimental methods of biophysics and/or biochemistry. Otherwise, the probability that we have reliable functional information on which molecular mechanisms are based decreases, reducing the predictive power of the interactomic models. PPIs are important for deciphering profound molecular mechanisms under normal or pathological conditions. Therefore, it is necessary to rely on curated databases with a high percentage of well-established binary relationships. I have used BioGRID and STRING, two very reliable public databases. They have undergone extensive integration, including the curation of thousands of journal articles, creating controlled vocabularies (ontologies) to describe PPI experiments, and defining standard formats for PPI data. They have introduced quality control measures by curation, methods for evaluating interactions, and approaches for linking interactions to context. The database BioGRID boasts one of the best average coverages (≈70%) of binary relationships, whereas about 79% of those in STRING have also experimental verifications.

The aim of this study is to differentiate the molecular effects of the S1 subunit in the human host when it acts together with other viral proteins and when it acts alone. I emphasize again that experimental methods are necessary to gain certainty about the interactions. To achieve this result, I applied a biological reverse engineering protocol. Reverse engineering is based on the direct validation of the biological message exchanged between two nodes of the net by validating it with external data. This involves deriving from an external reference model of the real biological relationships that exist between the nodes of the network without a priori knowledge of the computational protocols [18,19]. So, we can analyze the accurate mapping and prediction of protein interactions that contribute to disease pathology by exploiting PPINs. This approach can provide valuable insights into the molecular mechanisms underlying S1 action by mitigating low-resolution processes targeting this protein through a more systematic understanding of the complex regulatory networks in which it takes part [20,21]. However, this approach requires experimental validation of the interactions, which we achieved through BioGRID, in order to explain the molecular mechanisms underlying S1 actions. The results show the extreme functional complexity of the metabolic landscape in which S1 operates and how influential knowledge gaps and information biases become when we must evaluate where, when, and to what extent certain functional events should occur.

## 2. Materials and Methods

### 2.1. BioGRID

BioGRID (https://thebiogrid.org/) (accessed on 2 June 2024) is an important biomedical database that collects curated protein and genetic interactions, only from experimental studies and living cells [22]. Therefore, it represents a fundamental and unique resource for obtaining data on certified functional interactions in biological contexts. Through a specific Project (BioGRID COVID-19 Coronavirus Curation Project), BioGRID maintains complete and continuous coverage of protein interaction data between human proteins and all SARS-CoV-2 proteins. The Project is still active (https://thebiogrid.org/project/3) (accessed on 2 June 2024) and provides comprehensive datasets of curated direct interactions for the viral proteins encoded by SARS-CoV-2.

The dataset encompasses all experimental interactions between viral proteins and host cell proteins, as well as PTMs. I accessed the area SARS-CoV-2 Protein Interactions on 2 June 2024. In that area, we found all curated interactions between the virus and human proteome in 32 subgroups (the subgroup for ORF1a protein is void) for about 41,683 interactions with 25,620 unique interactors and information for 156 PTM sites on viral proteins. In particular, the protein S1 (GU280_gp02) shows 3031 curated physical interactions with 1371 interactors and 41 PTM sites (https://thebiogrid.org/4383848/table/severe-acute-respiratory-syndrome-coronavirus-2/s.html) (accessed on 2 June 2024) as supported by 903 publications.

BioGRID manages and integrates interaction data from low- and high-workflow experiments through a data curation and standardization process. This involves the analysis and validation of data from both types of experiments to ensure the quality and reliability of the information in the database. BioGRID considers molecular interactions detected through low-throughput experiments more significant than those found using high-throughput experiments. This is because low-throughput experiments are more targeted and accurate in identifying specific, biologically relevant interactions. In contrast, high-throughput experiments may produce a higher number of interactions but may also include interactions that are not biologically significant. I evaluate and integrate data from low-workflow experiments, which are considered more targeted and accurate in identifying specific and relevant interactions, into the database. BioGRID uses well-defined curation standards and data integration protocols to document and contextualize information from both types of experiments. This approach allows users to access a wide range of interaction data, from different experimental sources, in a consistent and reliable way. BioGRID contains over 2.7 million protein and genetic interactions, as well as over 1.5 million chemical interactions. The database is growing, adding new curated data.

### 2.2. STRING

STRING (Search Tool for the Retrieval of Interacting Genes/Proteins database) (https://string-db.org/) (accessed on 2 June 2024) Version 12.0 is a database of predicted interactions for different organisms [23,24]. The interactions include direct (physical) and indirect (functional) associations; they stem from computational prediction, from knowledge transfer between organisms, and from interactions aggregated from other (primary) databases. STRING is a database of known and predicted protein–protein interactions. The interactions include direct (physical) and indirect (functional) associations; they stem from computational prediction, from knowledge transfer between organisms, and from interactions aggregated from other (primary) databases. It considers conserved genomic neighborhood, gene fusion events, and co-occurrence of genes across genomes, as well as information about orthologs. STRING quantifies the strength of the evidence supporting each interaction by assigning it a confidence score. This score combines several sub-scores (based on seven channels of evidence), each of which is calculated in a personalized and source-specific way.

### 2.3. Protein Enrichment

Protein enrichment relies to some extent on prior knowledge, and statistical enrichment of annotated features may not be an intrinsic property of the input. To obtain a valid enrichment test from STRING, I input the entire set of enriched proteins into STRING, ensuring that “first shell” and “second shell” were both set to “none”. To confirm the procedure was correct, I also checked the STRING annotation, which disappears when the analysis is performed correctly. Next, I introduced new interaction partners to the network to expand the interaction neighborhood according to the desired confidence score. I used 0.9 as the confidence score. I always added 1st order proteins (direct interactions) first and then 2nd order proteins (indirect interactions), when necessary.

### 2.4. Cytoscape and Network Topology Analysis

Cytoscape [25,26] through Network Analyzer was used to analyze the topological parameters of networks. Using Cytoscape software (Version 3.10.1), I visualized and analyzed PPI networks, which offer diverse plugins for multiple analyses. Cytoscape represents PPI networks as graphs with nodes illustrating proteins and edges depicting associated interactions. I examined network architecture for topological parameters such as clustering coefficient, centralization, density, network diameter, and so on. My analysis included undirected edges for every network. I termed the number of connected neighbors of a node in a network as the degree of a node. P(k) was used to describe distributing node degrees, which counted the number of nodes with degree k where k = 0, 1, 2, … I calculated the power law of distribution of node degrees, which is one of the most crucial network topological characteristics. The coefficient R-squared value (R^2^), also known as the coefficient of determination, gives the proportion of variability in the dataset. I also examined other network parameters, including the distribution of various topological features. I calculated hub and bottleneck nodes based on relevant topological parameters. By examining the PPI network, I found the top 7 hub nodes. These nodes had higher degree values than the others and were in two central modules that were connected and compact.

### 2.5. CentiScaPe

Regarding centralities for undirected, directed, and weighted networks, CentiScaPe [27] calculates specific centrality parameters describing the network topology. These parameters facilitate users in locating the most important nodes within a complex network. The computation of the plugin produces both numerical and graphical results, facilitating the identification of key nodes even in extensive networks. Integrating network topological quantification with other numerical node attributes can provide relevant node identification and functional classification.

### 2.6. GO and KEGG Pathway Analyses

To better research and show the biological function of interacting proteins, I performed GO analysis, which included biological process (BP), cellular component (CC), molecular function (MF), and many other evaluations using the specific tools present in STRING. All functions shown by STRING are significant, having a *p*-value of <0.05.

### 2.7. SARS2-Human Proteome Interaction Database (SHPID)

I have collected in a single database all the files made available online by BioGRID, containing all the curated physical interactions of the 31 SARS-CoV-2 proteins gained through experiments in human cellular systems with viral baits, followed by purification and characterization with mass spectrometry [10]. These data are available as a zip file containing multiple zip files (32 zip files) each comprising interactions and post-translational modifications for each single SARS-CoV-2 protein for 33,823 interactions (as of June 2024). The database, therefore, contains the set of all real interactions existing between the SARS-CoV-2 proteome and all the proteins of the human proteome. However, some interactions could derive from artifacts of the method, such as non-biological interactions, because of the random encounter between proteins in the system used. All the interactions derived from BioGRID, even those with the lowest score, have a significant statistic with an FDR ≤ 0.01. This allows us to identify as many significant comparisons as possible while maintaining a low false positive rate of less than 1%. So only 338 interactions among all might be null. This database is the comprehensive repository of all interactions acknowledged as possible between the virus and its human host. The database also contains interactions between individual viral proteins, where known. As part of database search actions, one can ask who interacts with whom, with queries that use single human or viral proteins. The search can include multiple sets of proteins.

### 2.8. Highlighting the Nodes of a STRING Network Involved in the Same Biological Process (GO)

STRING makes visible the nodes involved in the same biological process, evidenced through its databases mapped onto the proteins (GO, KEGG, REACTOME, and so on), by activating the process line with a click of the cursor. Activation means that all nodes involved in the same metabolic process have the same color. Nodes involved in multiple processes receive multiple colors. This tool is very useful when one wants to analyze the involvement of multiple nodes in many metabolic processes, identifying which nodes represent the crossing points. Nodes that do not show any coloration identify components not involved (or influenced) in the activated process. The relationships that determine the coloring of nodes depend on the data and information extracted from the scientific literature in PubMed and from the databases connected to STRING. It also analyzes the quantitative impact of each data source, which contributes to forming the confidence score of each individual interaction.

### 2.9. Evaluation of the Hub-and-Spoke Model

Many properties of a scale-free network depend on the value of the exponent γ of the power law [28]. Therefore, it is interesting to establish how network properties vary with γ. Estimating the expected maximum degree (also known as the natural cut-off) for a scale-free network, which represents the expected size of the largest hub, is based on the following formula [29]:Kmax~Kmin 𝒩 1/γ^−1^(1)
where Kmax and Kmin are the expected maximum and minimum degrees for those nodes. 𝒩 is the system size in terms of the number of nodes.

### 2.10. Cluster Analysis

For the cluster analysis, I used the K-Means Clustering method [30]. K-Means Clustering is an Unsupervised Learning algorithm (centroid-based clustering algorithm) used by STRING to group the protein dataset into different functional clusters. Centroid-based algorithms are efficient, effective, simple, and sensitive to initial conditions and outliers. This makes it useful in handling networks. Here, for K, which defines the number of predefined clusters, I used the value of 10, which gave the most reliable clusters in terms of compactness, metabolic functionality, and *p*-value. There are several methods for determining K, the optimal number of clusters. I used the elbow method. It is a semi-empirical visual technique used to determine the optimal number of clusters (K) in a K-means clustering algorithm. It calculates the sum of the squared distances between points within each cluster and the cluster centroid (i.e., the sum of squared errors) for a range of values of K (e.g., from 1 to 10–15 or more). The plot shows K on the x-axis, with each y value representing the sum of squares for a specific K. At the optimal K value, the plot shows a curve concave upwards where the sums decrease, creating an elbow where the error becomes marginal. The value at the elbow represents the optimal value of the clusters to analyze. Adding further clusters does not lead to significant improvements. The K value of the elbow is a balance between the accuracy and complexity of the model.

### 2.11. Protein Intrinsic Disorder and Secondary Structure Prediction

I used the STRING feature and two online servers, Jpred 4 and IUPred2A. Jpred is a web server that takes protein sequences and, from these, predicts the location of secondary structures using a neural network called Jnet. It shows the prediction as a graph. IUPred2A [31,32] is a combined web interface that allows identifying disordered protein regions using IUPred2 and disordered binding regions using ANCHOR2. IUPred2A can identify disordered protein regions by analyzing their sequence, regardless of whether they are stable. Upon inspecting the graphic outputs of all the predictive systems, I have confirmed disordered segments in most of the examined proteins, whether viral or human.

### 2.12. Data Merging Methodology

Data Merging is a process in data management used to coalesce multiple related datasets into one. The Data Merging approach pools all data together and then estimates statistics on the resulting dataset of GO terms. The merging process enables the use of this combined data for more effective analysis, for extensive sets [33]. Data Merging merges disparate data sources, such as databases, or experiments data, into a unified dataset. I have used Excel (2016) for calculations. It aids in improving the accuracy of statistical data analysis, filling missing values in datasets, identifying correlations between variables, and making the data cleaning process more efficient. This procedure also presents some challenges. These include handling large datasets, ensuring the correct alignment of merged data, and dealing with ambiguities when datasets have similar identifiers. These issues, if not dealt with, can lead to data inconsistency or incorrect data interpretation. I have used this approach to integrate diverse data from various interactome analyses and data sources. The performance depends on the size of the datasets being merged and the computational resources available. With adequate resources, it is efficient and quick, providing a unified data view in little time. I have used a storage repository that holds a vast amount of raw data in its native format (Data Lake).

### 2.13. CIDER

CIDER is a web server developed by the Pappu Lab [34] for calculating parameters relating to disordered protein sequences, although it can generate values for any protein sequence. It is a Python backend, which allows you to run calculations, creating custom analytics pipelines. CIDER calculates a set of parameters which help translate primary sequence information into better understanding how the protein might behave, as well as produces a diagram of states [34,35,36].

The κ parameter is a parameter to describe charged amino acid mixing in a sequence. For a sequence of fixed composition, as κ goes from 0 to 1, we can think of the sequences as becoming less well mixed regarding the positive and negative residues. Useful parameters to combine with κ are the fraction of charged residues (FCRs) and net charge per residue (NCPR). As the fraction of charged residues increases, the relative impact of how those charges are spread across a sequence becomes more significant. They also relate to the conformational shape of the protein.

Disorder-promoting—we can categorize residues into disorder-promoting or order-promoting, as defined by Dunker and Uversky [37]. A ‘disorder promoting’ result reflects the weight of the fraction of residues, which forms the disorder-promoting set.

Regions on the diagram of states define the location on the diagram of states where the sequence lies. S1 lies in weak polyampholytes and polyelectrolytes, where sequences are globules and tadpoles (region 1).

## 3. Results

### 3.1. A Brief Analysis of the Behavior That We Expect for S1 Free in Solution

Chemical-physical data of the S1 subunit of SARS-CoV-2 Spike (Gene ID: 43740568 (ncbi.nlm.nih) and UniProtKB: PODTC2) were calculated by the web server CIDER [35], while their interpretation was exclusive to the author of this manuscript. The mature S1 subunit contains 711 aa, as decoded by its mRNA. We know many structural details of S1 in the Spike structure [38] and its mechanism of action in penetrating a living cell through ACE2 [24]. Spike (1273 aa in humans) is a precursor protein (Figure 1) that is proteolytically cleaved by furin into an N-terminal S1 subunit and the C-terminal hydrophobic S2 subunit. The latter mediates cell attachment. S1 is visible in the extracellular environment.

Its key role is to interact with ACE2 and, through its immunogenic epitopes, manage interactions with external proteins. The fate of the S1 particles released by furin has never received much attention [39]. However, S1 is detectable in the blood for a long time, both after infection and vaccination. The free structure of S1 at 3.6 Å is on PDB (7A91; entry DOI:10.2210/pdb7a91/pdb). Neutralizing antibodies target the epitopes on the S1 surface [40]. Thus, researchers fragmented S1 into peptides to understand which epitopes have the greatest antigenic power. They found a high number of strong epitopes in the 440–600 region, where the HLA epitope (448–456), called NF9 peptide, dominates [41]. What the researchers have overlooked is the conformational behavior in solution and the chemical-physical characteristics of the subunit, since they have studied techniques to develop detection assays [42]. These are important features for assessing the interaction tendency of proteins.

I used the platform CIDER of Pappu’s lab for calculations (see methods for details). We can predict with approximation its conformational preference as weak polyampholytes (FCR < 0.3) for S1 (FCR < 0.160), by combining chemical-physical parameters calculated by sequence [24,29]. The S1 subunit shows a fraction of charged residues (FCRs) of 0.160 and a net charge per residue (NCPR) of 0.006. This translates to a net positive charge at pH 7.0 of +4.95. The strong positive net charge of S1 at neutral pH favors interactions with negatively charged proteins or surfaces, but also suggests the excellent solubility of the protein in aqueous media. The FCR and NCPR values, as calculated by the CIDER server, suggest a dispersed positive charge distribution over the entire protein with a high number of charged patches in the region 400–600 and in the N-terminal tail. Also, a K value (charge patterning parameter) of 0.175 suggests segregation of charged residues within the protein with conformational fluctuations caused by long-range electrostatic attractions. This value is zero for sequences with well-mixed charges [34].

CIDER calculated a value of disorder-promoting capacity (DPC) of 0.549 [34]. The terminal segment, from residue 440 to about 700, is the one with the greatest number of disorder-promoting residues, according to Dunker [43]. This suggests that this region comprises many disordered segments. Flexibility analyses and structural models support this conclusion (see Appendix A [44,45,46] in Appendix A). The proline content of S1 (5.2%) is high (37 P residues, and, on average, one P every 19 residues). P is a residue that acts as a mobile hinge inducing structural orientation change, but it is also the protein residue with the most disorder-promoting potency [37]. We also need to put glycine on the same level as P because of its strong disorder-promoting ability [37]. Its content is also high, 46 residues (6.5%), with one G residue every 15.4 residues. This residue induces a strong structural flexibility in the structural environment that surrounds it, favoring broad structural fluctuations [47]. The set of highlighted parameters suggests an extended and mobile globule-like structure [34], very flexible with disordered segments but also very soluble in solution (see Appendix A) and susceptible to electrostatic interactions. This contradicts the general idea of compactness that arises from snapshot views of Spike by three-dimensional techniques. This extended flexible conformation benefits the virus by enhancing its ability to bind to receptors. The flexibility of the receptor-binding domain allows it to explore a wider space, increasing the likelihood of encountering a receptor and boosting its virulence. A fast equilibrium between protein–solvent and intra-protein interactions should control the conformational properties of IDR segments in solution, which are crucial for interactions [36,48].

These chemical-physical and structural properties of S1, free in solution, explain well the reason for the many interactions with human proteins found in BioGRID. But S1 also displays 87 sites for PTMs on its sequence. They modulate the structure and functionality of the protein in the different metabolic and temporal contexts in which it operates alone. Many sites in the IDRs undergo phosphorylation by serine/threonine kinases, which, by modifying the conformational properties of S1, allows it to coordinate many cellular signaling events [49]. These results highlight the importance of considering the intrinsic conformational behavior of this protein free in solution when developing vaccines because the final step releases S1 into the cell, free to interact with human proteins [50].

### 3.2. Data Source

All features highlighted in this study are based on experimental data extracted from BioGRID (see Methods, Section 2.1). I select 158 human proteins out of the 1371 unique interactors of S1, which induces 3002 raw interactions. The selected proteins are all characterized by a high significance level (level score ≥ 2) as well as by binding S1 with at least one Low Throughput (LT) interaction. BioGRID prioritizes molecular interactions detected with LT experiments over those detected with high-throughput (HT) experiments. This is because LT experiments are more targeted and accurate in identifying specific, biologically relevant interactions. HT experiments can produce a larger number of interactions, including interactions that are not significant [22].

Appendix A reports the proteins extracted from BioGRID that interact with S1 at LT. The Appendix A shows the interactome calculated by STRING for these proteins. The figure shows a compact network suggesting common functional activities. Although the confidence score is low (0.400) and all 7 channels are open to collect as much information as possible, the proteins form a compact network with an excellent *p*-value suggesting shared biological activities. The low score value and the use of all channels were used to collect as much information as possible, postponing the pruning of the less significant nodes to a later time. However, twelve nodes remained disconnected. The lack of connection suggests either little research on these proteins or that they are not involved in this specific functional context [51]. Therefore, I eliminated them so as not to alter calculating the topological parameters [52,53,54,55]. I pruned these nodes of low significance (CACNA1C, CLPTM1, CNTN1, CSNK1G3, IL1RAPL2, LYSMD3, MPZL1, MSMP, PIM2, SLC6A15, SLC7A4, and WDR45B) to increase the accuracy and robustness of my conclusions. In Appendix A, I show the new 146-node interactome. This new interactome appears well organized, with a central compact body and many peripheral subgraphs, locations of specific biological activities. For instance, the subgraph on the left (ZDHHCXX-GOLGA7) is the palmitoyl–transferase complex involved in protein transport from the Golgi to the cell surface [56]. The pentagonal subgraph (bottom compared to the previous one) shows components of the Coatomer cytosolic protein complex II (COPII), which promotes forming transport vesicles from the endoplasmic reticulum (ER) and regulates the intracellular membrane trafficking, from the formation of transport vesicles to their fusion with membranes [57]. Many of these 146 nodes have all the characteristics of functional compactness and high rank to maximize the metabolic processes of the network through enrichment as functional seeds [58]. Efficient seed selection should select the most influential nodes to achieve the maximum level of functional influence. This is because, in the enrichment phase, the robustness of seeds is essential to counteract potential disturbances, such as topological alterations. An accurate selection of influential seeds reduces perturbations in the network [59].

Functional enrichment is based on statistical parameters related to biological functions associated with the gene set extracted from BioGRID. It identifies biological and functional themes (pathways, Gene Ontology, diseases, etc.) that, although sometimes over-represented, apply to the topic under study. Integrating multiple pathways (KEGG, Reactome, etc.) offers advantages in terms of more probable, more extensive, and robust functional annotations, necessary for a better understanding of the functions and metabolic regulation existing in a complex biological system such as the virus–host one.

I have two major goals: extracting useful information from the functional processes of the proteome that are related to functional seeds, as a strategy; defining the topological space in which to represent and visualize the structural organization of the extracted metabolic processes as a method. STRING implemented the calculation for the functional enrichment of these nodes by adding 500 first and 500 second order proteins (direct and secondary interactions) until obtaining an interactome of 1146 nodes (see Appendix A). This interactome, despite being very compact and with an excellent statistic, still has disconnected nodes, most likely because of heterogeneous data. Although access to scientific documents in natural language by Text Mining is easier, the results of this automated search are often not relevant to the needs of the user looking for experimental and quantitative data [60,61]. In fact, extracting information through key phrases and relationships used by these systems leads to heterogeneous results with differences among the scientific databases from which the articles were retrieved, even if articles are similar [60]. It is important to note that bioinformatic platforms treat less studied genes/proteins as if they were background noise and often eliminate them from the calculations [62,63]. This generates uncertain in predictions or information, so I eliminated disconnected nodes. The Appendix A reports the pruning protocol with the degree-lists of the 1060 residual nodes and with the 87 nodes eliminated. Figure 2 shows this interactome (from now on “interactome-1060”) as calculated by STRING.

### 3.3. Interactome-1060

Figure 2 shows an interactome comprising proteins from the human proteome selected through a sequential selective process that identified those with the highest experimental probability of being involved in the metabolic processes induced by the S1 protein in the human organism. The interactome comprises 1060 nodes and 17,494 edges, obtained by selecting the highest significance interactions (confidence score 0.900) and excluding the Text Mining channel. This is another important point because of the uncertainty arising from detections of protein interactions, which is reflected in the network’s structure [51]. PIN (Protein Interaction Network) analysis should be reproducible, by similar results across different scoring thresholds of calculation systems. This suggests that, for maximum confidence, we need to have a robust metric across the network to have meaningful and reproducible topological results [51]. The topology of interactome-1060 is complex because of the many peripheral subgraphs enveloping an extended central core. Specific nodes at the interface connect the various peripheral subgraphs through a few interactions with the central body and with each other. We can observe subgraphs (also called communities or molecular modules) densely connected within themselves but poorly connected with the rest of the network. The intensity of the connections and the compactness of each subgraph suggest they represent molecular complexes that carry out specific and common functional activities [64,65,66,67]. This broad functional connectivity shows the possibility of an extensive repertoire of responses to stimuli. After all, the cell is a complex multi-agent system programmed to perform predefined functions at specific times. Therefore, interactome-1060 represents a robust set of human proteins suitable for a reverse engineering approach. With it, we can assess the significance of each single interaction by evaluating its real biological meaning. I believe that this approach has a broader value than the rather reductionist meaning of reverse engineering as a technology [67]. In short, I try to discover the one-to-one interactions of S1 in the network by validating them through external biological information.

#### 3.3.1. Quantitative Aspects of Interactome-1060’s Functional Processes

Table 1 shows an overview of the functional processes activated by interactome-1060.

The control of so many Biological Processes by S1 is remarkable. This analysis is based only on experimental data from BioGRID, functionally analyzed by STRING. The calculated interactome is based on selecting only the most statistically significant interactions (through the highest confidence score) deriving from data and information from over 10,000 scientific articles and from eliminating information from the Text Mining channel which introduces biases in the data and in information compared to every other approach. Such an approach favors the greatest possible certainty of the interactions in the interactome (Figure 2). However, the more complex a network is with many multi-node interactions, the more intrinsically robust it is with reduced false positive interactions. The Appendix A, reports the node degrees of interactome-1060 entirely. The node with the highest connectivity is RPS27A (nodal degree: 230), a ribosomal protein. Hubs connect multiple nodes to centralize network traffic through a single connection point. Barabasi suggests that the range of degrees for including the HUB nodes should be half the value of this node [29,67,68]. This range includes 65 HUB nodes out of 1060 nodes (6%) from 230 to 115 degrees. A closer inspection shows that these nodes are almost all ribosomal proteins, even if in different roles.

Among these high-ranking nodes, four of them regulate and control many ribosomal activities, showing more interactions than other proteins. They are RPS27A, its paralogue UBA52, FAU, and RACK1. RPS27A and UBA52 play crucial roles in targeting cellular proteins for degradation by the 26S proteasome, maintaining the chromatin structure, and regulating gene expression and the stress response [69,70]. FAU is a protein contributing to the assembly and functionality of 40S ribosomal subunits in the cytoplasm [71]. It plays a role in ribosomal biogenesis and is associated with various protein complexes, contributing to regulating the cell cycle. RACK1 is a protein that controls translation and acts as a scaffold for signaling to and from the ribosome [72]. Upon viral infection, RACK1 remodels ribosomes so that they become optimal for translating viral mRNAs but not host mRNAs [73]. Thus, they interface with multiple cellular functions and processes. Here, I focused on their pivotal roles in the synthesis of new proteins. To gain more insight into their activity, I used the STRING action “recenter” that rewires the network around these proteins, showing all the proteins in STRING that interact with them. This specific interactome (Appendix A) reveals a strong connection between the four proteins and their control over the remaining 793 proteins. The functional picture that emerges is that of four essential cytosolic small ribosomal subunits involved in viral mRNA translation (GO:0002181 Cytoplasmic translation, *p*-value: 4.16 × 10^−85^; GO:0042274 Ribosomal small subunit biogenesis, *p*-value: 3.79 × 10^−45^; GO:0006412 Translation, *p*-value: 2.67 × 10^−194^; GO:0006364 rRNA processing, *p*-value: 2.85 × 10^−88^; GO:0042254 Ribosome biogenesis, *p*-value:1.38 × 10^−109^; GO:0022613 Ribonucleoprotein complex biogenesis, *p*-value: 9.71 × 10^−126^; GO:0034660 ncRNA metabolic process, *p*-value:1.86 × 10^−65^; CL:143 Viral mRNA Translation, and Sec61 translocon complex, *p*-value:1.26 × 10^−91^; HSA-192823 Viral mRNA Translation, *p*-value: 2.39 × 10^−72^). All this shows a dynamic ribosome action in mediating crucial cellular mechanisms, even in pathologic states. It is a view quite like that of some authors who contest the traditional view of ribosomes as static and invariable entities [74,75]. To support this consideration, studies have shown that certain ribosomal proteins impede viral action in cultured human cells, leading to changes in human functionalities [76,77,78].

#### 3.3.2. Significant Topological Parameters of Interactome-1060

Regardless of the deep molecular machinery underlying the functional characteristics, the space that emerges from the analysis of these topological configurations provides a logical substrate for understanding viral strategies. The main topological characteristics of this interactome (see Table 2) reveal important principles of cellular organization and functionality. The extensive eccentricity of the network, as shown by the high values of its diameter and radius (10 and 5), suggests functional peripheral subgraphs (or communities). The heterogeneity (1.187) supports a large tendency to have hub nodes [79], while a centralization value close to zero (0.189) supports compact and dense connections within the network. Another interesting parameter is the value of the average clustering coefficient (0 ≤ C ≤ 1), which reflects a modular organization [80] that, in light of the large diameter, also suggests an asymmetric architecture, as we observe it.

#### 3.3.3. The Power Law of Interactome-1060

However, before any topological consideration, it is necessary to find out what distribution the interactome degrees show. Biological networks show scale-free behavior with a few hub nodes controlling multiple connections within the system. The lack of an internal scale means that nodes with a large difference of degrees coexist in the same network. Barabasi observed this feature in many biological networks [83,84], where the fraction of nodes with degree k follows a power law distribution, revealing that the degree distribution of the network is well approximated by Pk~k^−ƴ^, (ƴ > 1). The exponent ƴ is the degree exponent, and many properties of a scale-free network depend on the value of the degree exponent [83,84]. Calculating the degree distribution is an important part of analyzing the properties of a network [85]. Figure 3 shows that interactome-1060 follows the characteristic distribution law of the nodes of a scale-free network.

To test the distribution, I fitted the function f(x) = ax^b^, where the values of a, b (degree exponent), and R^2^ are 181.8, −0.98, and −0.272. Even though interactome-1060 shows a significant *p*-value of 1.0 × 10^−16^, the low correlation index of this fit underscores a strong expectation of heterogeneous associations among nodes, such as high clustering. The presence of clusters in the network topological architecture is useful for defining in a non-random way specific pharmacological attack points [86]. We can note that nodes with high connectivity form a long tail (long-tailed distribution) and between degrees 30 and 70, there is a peak that characterizes an excessive number of nodes with these degrees compared to the fit. Some protein networks acknowledge the long tail distribution as an intrinsic property rather than a byproduct of the specific algorithm used to compute the network [87]. This is also a characteristic property of scale-free networks that result in distributions with long tails where only the terminal nodes have high degree values [88].

In most real-world networks, new nodes prefer to connect to more connected nodes, according to a process called preferential attachment [89]. Therefore, the number of nodes grows because of the addition of new nodes, so growth and preferential attachment coexist. The power law should represent this tendency of the nodes. If we examine the degree distribution in the log–log graph (inset of Figure 3), we find that the distribution deviates from a pure power law, which in logarithmic representation should follow a linear trend. The log–log distribution shows many overlapping linear plateaus in the high-k regime (the long-tail nodes) and a clear distortion in the low-K data. This suggests various subgraphs (molecular modules), each with its own specific hubs [90].

According to the Barabasi’s model [83,84], for b < 2, the exponent ƴ will be larger than one. Hence, the number of connections to the largest hub will grow faster up to reaching the global size of the network. This means that for a very large N (total number of nodes), the highest degree hub could gain all nodes in the network. But this tendency slows down the connection speed [90,91], allowing other nodes to increase their connectivity. Barabasi unraveled the complexity of this phenomenon, showing how a large scale-free network with b < 2 cannot exist without multi-link subgraphs [83,84]. Networks are not a static entity, but grow by adding new nodes. The joint necessity of growth and preferential attachment generates scale-free networks and changing either of these factors will cause changes to the scale-free properties and network topology. Therefore, the growth rate of a hub node depends not only on its age, because other nodes can transform random transient interactions into a long-lived interaction. A common characteristic of these last nodes is an intrinsic property that we will call fitness [92,93]. Fitness is a property that favors the preferential attachment to other nodes by increasing the growth rate of their connectivity [94,95]. It is based on the set of distinctive structural and/or functional characteristics possessed by each node. On this basis, Barabasi has developed a specific model, the “Bianconi-Barabasi Model” or “Fitness Model” [92,96]. This model shows how nodes with different internal characteristics can gain links at different rates. It predicts that the growth rate of a node is determined by its fitness. One can measure the fitness by comparing the node with the temporal evolution of the fitness of other nodes in the network. This model presents a behavior of the nodes that is like that of Bose gas, studied by physicists [97].

This similarity explains very well the physical basis of forming the many independent and dense functional subgraphs observed in protein networks, characterized by their hubs. In fitness distributions, the network exhibits a “fit-get-rich” dynamic, meaning that the degree of each node is determined by its fitness, where new links not only arrive with new nodes but also occur between pre-existing nodes. The fitness model also shows that in many real systems, nodes, and links can change and disappear, explaining why nodes disconnect after enrichment and therefore need to be deleted [98].

However, if the linear preferential attachment governs the growing network, then a pure power law should emerge. However, it is rare to observe a pure power law in actual systems [99]. The Barabasi–Albert model is an idealized model that represents only the starting point for understanding the distribution degree in real networks [100] because fitness plays an important role. The concept of fitness in protein networks refers to the ability of a protein entity to survive and thrive within a protein network, because of its interaction with other proteins and its functional relevance. This plays an important role in forming those protein complexes that are crucial for a variety of Biological Processes. Protein fitness is based on many essential protein properties, such as secondary structure, solubility, binding affinity, flexibility, and functional specificity [101,102,103,104]. Therefore, in Interactomics, we can motivate this model only if there are experimental observations that explain the internal characteristics of the nodes. If we identify them correctly, then we can understand how fitness contributes to forming subgraphs and the topological evolution of the network [105]. In networks with many subgraphs, such as the ones in this study, hubs connect to nodes of a small degree. As a result, we have a network that is unlikely to be represented by a single giant component. Networks in which hubs avoid connecting to each other but connect to many low-degree nodes are called disassortative and generate a hub organization with a hub-and-spoke pattern [106]. The logarithmic representation of the disassortative degree distribution is characteristic and very similar to the one calculated in Figure 2. This means that the network (Figure 1) has intrinsic difficulty being represented as a single giant component. We can appreciate this feature by measuring the slope of the linearization of its distribution. As we will see later, it is a measure of the speed of growth. The fit shows a negative slope (see Figure 2 caption), and a lower probability of mutual interaction (y-axis) characterizes the nodes with the highest rank (x-axis) [107]. In conclusion, many significant subgraphs give us the picture of the fundamental functions that the biological system performs and its dynamics. This allows us to understand with certainty the behavior of S1 in the system.

#### 3.3.4. Origin of the Node Fitness in Interactome-1060

A question now arises: which structural property dominates the fitness of interactome-1060? Certainly, each single protein-node has its own specific intrinsic properties, but which of these is the predominant one? I have considered many characteristics of the nodes (protein length, secondary structure, flexibility, intrinsic disorder), but one of them stands out above all. Protein–protein interactions within a cell are dynamic events which do not occur concurrently and in the same location. When molecules come together, they form complexes, where the structural disorder of motifs at the interface often mediates transient interactions. This creates transient multi-state complexes that are characterized by dynamic assortments of subunits. These diverse transient combinations also facilitate the occurrence of different entropically driven conformational states [108]. Thus, the intrinsic disorder is the most critical feature associated with transient interactions. Weakly interacting proteins show a fast dynamic bound-unbound equilibrium, which also includes interactions that are triggered by an effector molecule and stabilized by a conformational change. The 66% of signaling proteins involved in cellular functions with strong temporal variation of activity show a high probability of involving transient interactions [109]. Temporary interactions wield a significant influence in determining the hub behaviors [110]. Many hub proteins with variable co-expression partners show transient binding at different times. With high co-expression partners, they develop stable complexes [111]. This highlights the importance of the intrinsic structural disorder in protein–protein networks, but also underlies connectivity and controls how hub proteins interact. Figure 4 shows the average distribution of the intrinsic disorder existing in the interactome-1060 interactions.

The plot shows that intrinsically disordered proteins, which have a disorder percentage greater than 30% and play a crucial role in interactions, account for about half of the total interactions. The disorder content is very large, but almost all proteins show disorder. However, disordered segments can make interactions even shorten. All high-ranked nodes have a disorder content between 20 and 40% [112]. STRING used the binned pseudo-R-squared (BP-R^2^), a measure developed by Lun et al. to quantify complex signaling relationships between two variables, to assess the goodness of the fit [113]. The idea is to capture relationships that may not have a high Pearson or classification correlation, but that show associations which are non-trivial. The range between 0 and 70 degrees shows a concentration of proteins with a high intrinsic disorder content. This range contains many of the proteins with high fitness potential. Disorder-enriched hub and non-hub nodes also show a higher number of links, because of the higher number of targeting, catalytic, and many types of PTM sites [114]. In conclusion, we can say that protein disorder is prevalent among these proteins. This is the structural feature that dominates fitness, driving the connectivity within interactome-1060.

#### 3.3.5. Centralities-Based Analysis of Interactome-1060

Topological analyses applied to protein networks show that some parameters such as connectivity degree (k), betweenness centrality (BC), closeness centrality (CC), eigenvector centrality (EC), and eccentricity are crucial parameters of nodes [115]. They are indicators of centrality because, when assigning rankings to nodes within the graph; they characterize the most important vertices. Each centrality measure assigns a centrality value to each node in a network and captures different aspects of what it means for a node to be important in that graph. High-ranking target search, identifying suitable nodes for characterization, is a critical step in annotating functional processes and understanding their molecular basis. The priority is to narrow down the most important nodes. After defining the broad interactome induced by S1 (interactome-1060) and calculating its properties, it is useful to identify hubs and bottlenecks [116].

We can define as hubs the top 10% of the nodes in the high-confidence protein interactome based on their node degree (the number of interactions associated with a node). We can consider another top 10% of the nodes, ranked by betweenness centrality and closeness centrality (BCC), as bottlenecks. Betweenness centrality is an indicator of a node’s centrality in a network. It is equal to the number of the shortest paths from all vertices to all others that pass through that node. Closeness centrality calculates the average distance of all the shortest paths between a node and every other node within a network. Thus, nodes with a high closeness score have the shortest distances to all other nodes. Betweenness and closeness are a way of detecting bottleneck nodes that can spread information through a graph, even if they do not always have very high degrees [117,118,119]. The eigenvector centrality of a graph is a measure of the influence of a node in a connected network. Each node in the network receives a relative score, acknowledging that connections to nodes with higher scores contribute more significantly to a node’s own score than connections to nodes with lower scores. A high eigenvector score shows a node has connections to many nodes that have a high score themselves. The resulting information allows us to identify key nodes in terms of connectivity relevance in the interactome, thus suggesting very similar and super-imposable results to those of node degree [120,121]. We know that nodes with large k are central because they might correspond to disease-causing genes/proteins, whereas bottleneck nodes are vital since they serve as a crossroads in major signaling “highways” or overpass across these “highways”. Therefore, I focused on the hubs and bottlenecks that were central to the PPI network, identifying these key proteins and considering their sub-networks as the backbone of S1-induced topology [122,123].

Figure 5 reveals the node distributions according to their centralities as calculated by Cytoscape. I report the protein names for comparing them with the results in Table 3. I extracted the 26 nodes with the highest centrality values from each distribution. They represent the candidates for the analysis of the topological properties. Both closeness and betweenness select the nodes with the best features as bottlenecks. The eigenvector distribution shows the protein nodes with the highest connectivity comparable to high-degree hubs. We can find the 26 values for each centrality in the Appendix A. In the comparisons between the various sets (betweenness vs. closeness and eigenvectors vs. degree), I selected only nodes in common between both compared sets. This resulted in fewer bottlenecks. But even though the total number of selected nodes was lower, I created very significant sets of hubs and bottlenecks (Table 3).

I computed hub and bottleneck nodes based on relevant topological parameters, but there is no consensus in the literature on a defined threshold to identify how many nodes should be hubs or bottlenecks in a protein network, because the possible and used criteria are too numerous and sometimes arbitrary [123]. However, these nodes should represent the backbone of the basic connectivity that should favor a balanced architecture of the entire network with specific functional aspects. Therefore, I have gathered the selected hub and bottleneck nodes in a group of 33 nodes (3.1%). This group (Table 3), besides the topological characteristics, should also show evidence of reliable interactions, forcing the network into a hub-and-spoke organization. We should expect that, for the topological functions they perform, they should connect to each other, because they all interact with the same dominant hub. In this regime, hub-and-spoke configurations show characteristics of disassortative networks [29,67].

A look at the two plots in Figure 6 reveals that many of these high-ranked nodes (hubs and bottlenecks) concentrate in the nucleus and cytoplasmic system. This agrees with all activities related to viral translation. Both categories of nodes have significant biological importance, as they represent key connections and critical points of interaction. Bottleneck nodes possess a high centrality of intermediation, as they connect many parts of the network, influencing the information flow. The high scores between 4.5 and 5 justify these attributions.

The hub-spoke architectural pattern (for an operational explanation, see https://cloud.google.com/architecture/deploy-hub-spoke-vpc-network-topology) (accessed on 2 June 2024) expects the core system with high connectivity to comprise hub nodes, while the well-connected bottleneck nodes are located outside. Here (graph on the left), some bottleneck nodes (UBA52, MED1, RPS27A, RACK1, CD74, FAU, EF1A1) operate at the interface linking the remaining nodes (SRC, ACTB, STAT3, CBL, CD44, EGFR). In the graph, the colors identify the main overall functions they manage together. The red color shows cytoplasmic activities while the blue refers to overall nuclear ones. As we can note, all hub nodes operate in both compartments. Obviously, the graph highlights only the direct connectivity between these nodes because, in the complete network, each hub, or bottleneck node manages many other “normal” nodes. White bottlenecks mediate many signaling activities. ACTB is involved in cytoskeletal control (GO: 0005925, focal adhesion, *p*-value 4.07 × 10^−21^).

### 3.4. Justifications for a One-to-One Study

The principles governing immune responses operate at the organ scale. The propagation of immune signaling between organs shows inter-organ mechanisms of protective immunity mediated by soluble and cellular factors [124] that transcend organ boundaries [125]. Cellular factors such as memory T cells can patrol organs and infected tissue [126,127]. Changes in tissue gene expression following vaccination have also highlighted immune processes that operate at the organ scale through a protective network [124]. Recent work shows that vaccination, like repeated infections, provides protection even in very distant tissues [128,129]. Researchers have used this logic to study shared and tissue-specific expression patterns [130,131] and their correlation with disease [132,133]. All this suggests isolating the biological activities one by one, specific to the S1 protein. Both during infection and vaccination, both events have in common the encoded information (mRNA). Therefore, in both cases, the mRNA must use the same biosynthesizing nano-machines, and the decoded protein, when acting alone, should take part in the same cellular processes present in the human host. In Appendix A, I provide the structural meaning of the one-to-one interaction used in this article.

### 3.5. Reverse Engineering

Reverse engineering in biology applies an engineering concept, that of dismantling a process to understand it and discover the biological strategy. Thus, it is often used to discover the design principles of a biological system when the relationships between microscopic and higher-level processes are degenerate (many-to-many or one-to-many). It addresses the understanding of a complex system when the non-linear relationships between the system’s capabilities and its deep molecular mechanisms change. This suggests its usefulness in analyzing a complex functional system, faced with limited a priori knowledge of its “design principles” [134]. At first glance, “disassembling to reassemble” may seem like a reductionist approach to systems biology. However, data-intensive biological fields use reverse engineering approaches to recognize nonrandom connectivity patterns and identify the functional capabilities of the overall network architecture [20]. This enables a topological analysis that abstracts from the context of network connectivity to identify functional capabilities. It is an approach to understanding how certain components are wired to create a functional whole. The search for these design principles allows us to know lower-level causal details and becomes robust when integrated with external experimental data tested in vivo and which can therefore biologically validate an interaction as real [135].

My goal is to understand whether, in the same system but with changed organizational features, S1 performs similar operations Although specific metabolic parameters influence the activation of molecular mechanisms and functions, we might identify parameter spaces that support the same functions. In a broader discussion, we should consider that groups of viral proteins contribute to enhance the virulence of the virus by attacking single human proteins with multiple interactions, but some also do so alone. This should not be surprising because I have already observed it in the liver affected by COVID-19 [10]. SARS-CoV-2 shows a broad tissue tropism, although of varying degrees, perhaps greater than what clinicians can appreciate through observation. This tropism is, however, expressing the steps necessary to progress the viral infection, even in phenotypically different individuals, and represents a strategic adaptation to the host. Achieving success in replication requires the virus to leverage Spike’s interactivity and adapt its proteome strategy to the host’s unique metabolic landscape. Among the main variables that the virus encounters are age, sex, nutritional status, and previous individual pathologies.

In the Appendix A, I show the overall results of the reverse engineering analysis. I checked, one by one, all 1060 nodes of the interactome in Figure 1 against the 25,521 interactions collected in the SHPID database [10]. These interactions derive from individual proteins encoded by the SARS-CoV-2 proteome and those of the human proteome, as reported in BioGRID.

This file shows that there are multiple interactions of S1 together with other viral proteins (Appendix A). I also found many viral proteins that interact in a one-to-one manner with specific human proteins (Appendix A). They could be pharmacological targets. Many human proteins are not involved in any viral activities. This result confirms my previous observations on COVID-19 [10]. Proteins not involved control metabolic processes that are beneficial for both the virus and the human host. The most interesting observation (Appendix A) is a set of 27 unique one-to-one interactions of S1 with human proteins (ACE2, AGTR1, AKT2, APOE, ASGR1, AVPR1B, C1QB, C1QC, CD46, CFH, CFP, CLEC4M, COP1, CR2, DPP4, ESR1, F10, FLT1, L12RB1, ITGB6, LYPLA2, MBL2, NID1, SDC1, SDC2, SNCA, TLR4). Through these proteins, we can try to understand in which functional processes they are involved, with which human proteins, and whether these interactions could represent a functional framework exclusive to S1, or to the infection.

The multiple interactions refer to the attacks conducted by groups of viral proteins, including S1, against single human proteins. The file (Appendix A) lists 148 human proteins attacked in this way. A careful observation shows that the number of viral proteins attacking single human proteins is often considerable. An example of this, the gene EIF2S1, Eukaryotic Translation Initiation Factor 2 Subunit Alpha, encodes the protein IF2A_Human. This is a protein of 315 aa, with a mixed alpha/beta structure. It is a member of the eIF2 complex that functions in the early stages of protein synthesis by forming a ternary complex with GTP and initiator tRNA. Nineteen viral proteins (nsp1, nsp3, nsp4, nsp5, nsp6, nsp13, nsp14, E, M, N, S, ORF10, ORF3a, ORF3b, ORF6, ORF7b, ORF7a, ORF8, ORF9b) attack this protein. Given its small size, like that of the ternary complex itself, it is impossible for there to be enough surface space for the interactions of 19 proteins concurrently. This means that the interactions, although all brief and momentary, occur at different times. However, without the time sequence, it is impossible to define the actual functional mechanism affected by these interactions (even without wanting to consider the “where”). On this basis, it once again seems useless to define an overall mechanism through chronologically undefined single interactions.

### 3.6. Interactome-814

I used these twenty-seven proteins as functional seeds in the human proteome. Figure 7 shows the new interactome calculated by STRING.

This interactome (from now on “interactome-814”) comprises 814 nodes (Appendix A). The first observation is that despite that I added 1000 proteins for the enrichment, the system only accommodates 787 of them (814 − 27 = 787). This seems to reflect a low number of experimentally proven interactions. We can consider that STRING classifies only 21.46% of them as High or Highest (Appendix A), which brings us back to the considerations made in Appendix B. Table 4 shows that this interactome too has a periphery rich in subgraphs, but is on average less dense (0.22), with a value of the average number of neighbours about 50% lower than interactome-1060. Heterogeneity (1.042) suggests the tendency of this network to contain hub nodes, while the centralization value (0.138) still supports compactness, even if the distance between two nodes (diameter) is lower but still high, and supports the almost asymmetrical architecture we observe. In conclusion, we have an interactome with a global organization quite like the previous one, although smaller and less dense in terms of connectivity.

Also, interactome-814 shows a power law characteristic of scale-free networks (Figure 8). Differently from interactome-1060, the log–log distribution plot shows a fit with a good R^2^ value of 0.7528, so this log–log fit is the signature of a system well described by the power law equation. Hence, interactome-814 should show a very balanced and linear overall growth, without distorting effects. Also, in this case, the exponent y is greater than 1, showing a central component that does not prevent peripheral modules. As for the two slopes, comparing them, they are both negative and not very different in value. However, the slope shows different growth rates, with the number of nodes increasing faster in 1060. The two interactomes, although similar, react differently to internal or external factors, and this could be because of the greater heterogeneity found in 1060. All this suggests that, despite the considerable underlying biological complexity, the relationships between metabolic processes and population sizes of the interactomes seem to obey a simple relationship, given by the power equation. This is a further fact that justifies the comparisons I am making of the two interactomes and also the search for the specific functional activities of the S1 protein.

In Figure 9, I show the centrality distributions of interactome-814. I reported the numerical values of the first 26 terms for each distribution in the Appendix A. The same procedure adopted for interactome-1060 was used to assign the highest-ranking values. We can see in the betweenness centrality distribution that the upper range of the distribution is very wide, involving proteins with both a high degree and medium-low degree. What is striking is that some of them are also present in the centrality distribution of the eigenvectors. Since these are different topological properties, this, as we will see later, suggests mixed proteins (hub/bottleneck), a situation not present in interactome-1060. 

Table 5 reports the results showing the highest-ranking hub and bottleneck nodes. A comparison with Table 3 shows that although the architecture of the two interactomes may seem quite similar, the main proteins that underlie their structural and functional organization are different and behave differently. The individual nodes in Table 3 perform only one activity, either as a hub or as a bottleneck. There are no mixed-activity nodes. We can consider those in Table 3 as pure hub and bottleneck nodes [136], while many of those in Table 5 show mixed activity.

The functional coincidence between some hubs and bottlenecks in the interactome shows that these proteins not only cover many interactions but play a critical role in maintaining connectivity and stability in the network [137]. The coincidence also suggests that these proteins are fundamental to the function of the biological system and may represent key points for therapeutic interventions or functional analyses [136]. In fact, both categories of strategic positions in the network help to understand the robustness and vulnerability of the interactome, revealing potential regulatory mechanisms [138]. This allows us to consider that the S1 subunit behaves differently when it interacts alone with the human proteome. To obtain a more reliable picture, I verified whether a hub-spoke scheme also exists in this case and the main allocations these proteins have in the cellular compartments. Figure 10 shows a hub-spoke scheme where the central system is mixed because both pure hubs and bottlenecks man it.

The processes shown in the table are just some of the most relevant terms in which the nodes of the hub-and-spoke organization are involved. The graph shown is the structural backbone of the network. These activities, as well as many others not reported, support the deep involvement of S1 in metabolic activities, even with worrying negative aspects (hsa05200 or HAS-199418). All processes show high strength values, which suggest coordinated and active processes, and are well supported even at the gene expression level. The graph includes 23 nodes among the highest-level ones, and 22 of them are involved in well-supported and significant negative processes.

Figure 11 shows the four most significant distributions relative to the cellular compartments (cytosol and nucleus) and tissues (nervous system and blood) populated by the proteins of interactome-814. The upper parts of the distributions exhibit dense populations between the values 4 and 5. This shows the high functional activity of the proteins that populate it. The extent of involvement of high-ranking proteins can be determined by analyzing the distribution along the abscissa (degree). In this interactome, the cytosol, and nucleus stand out as the most involved and populated cellular compartments. However, the extracellular area and membrane level also exhibit intense metabolic activity. Among the tissues, the nervous system is involved by proteins that include many of the high-ranking ones.

In summary, the two interactomes, despite their similar structure, perform distinct functions that are only broadly defined. It becomes important to focus on the functional activity to understand if, and how much, they differ from the point of view of metabolic purposes and, above all, which genes oversee these processes. Are they the same genes or are they different genes? What is surprising is that interactome-814, despite having a lower total number of nodes than interactome-1060, controls 7120 terms and 40% more functions based on Gene Ontology terms (see Table 6). Appendix A summarizes the major functional roles of interactome-814. I highlight the major subgraphs, showing one of their primary functions. The Data Merging approach will implement and detail this still-rough functional summary later.

In Barabasi–Albert network models, enrichment arises from a network growth process governed by the preferential attachment of nodes. The same protein can exert different functions by binding to different partners. A fundamental question is to understand how the opportunistic choices of individual nodes shape the properties of the global network. Identifying these influential nodes is a challenging and still understudied task. We also have to consider that nodes are biological agents and links represent their functional interactions, which can also be modeled as cooperative activities. Nodes, taking part in an ever-increasing number of molecular processes, can change their local behavior or topology, maximizing their cooperative activity [139,140].

How does a protein select among a multitude of potential binding partners within a cell, expanding its functional repertoire? An adequate response should consider the location and translation rate of messenger RNA (mRNA), as both factors can cause spatial regulation of protein synthesis, affecting local protein concentrations and interactions [141,142]. The rate of translation elongation can indeed influence protein folding and its interactions with other proteins [143,144]. Slow translation can allow more time for co-translational folding and interaction with certain partners, while rapid translation might favor interactions with different proteins or lead to misfolding [145,146]. However, additional considerations can also come from other types of comparisons of the two interactomes.

### 3.7. Data Merging

The two interactomes, 1060 and 814, although induced by the same viral protein, appear to operate in different metabolic contexts. Characterizing the behavior of these two networks is essential to understand the complexity of S1 action [147]. The differences appear clear if we compare the set of GO processes controlling each interactome. The enrichment of interactome-814 shows 7120 terms in 15 categories. Interactome-1060 shows 4989 terms in 15 categories. The difference in terms is 1.42-fold, but for ontological terms, it is 40%, and the three Ontologies reflect the functions. The size and reliability of the datasets under study, the scientific design, and the phenotype specificity affect identifying critical nodes and functional processes in any system. I standardized these variables by making the methodological procedures as similar as possible and, most importantly, using only experimental data and selecting only those with the highest reliability. I considered the topological properties of nodes and evaluated their functional roles based on their ability to transmit information within and between modules in the network. Using Gene Ontology for genomic functional annotation is crucial, as it can reveal important biological information. Gene Ontology (GO) comprises three categories: molecular function, cellular component, and biological process [148]. But it has redundancy problems when analyzing them together, especially because of gene overlaps. The redundancy in GO annotations can complicate interpreting biological data [149]. Therefore, the analysis of a single ontology, such as Biological Processes, which are also the most abundant and all-encompassing, can be a useful strategy to limit the redundancy and improve the clarity and significance of the results [150,151].

By comparing the Biological Processes (GO) of the two interactomes, I still highlighted the large functional differences already noted. There are 2,557 processes for interactome-814 versus 1430 processes for interactome-1060, which is 44.1% more. A closer look at the two interactomes (see Appendix A) shows that many functions are similar, while others appear specific to each of them. The same happens for many of the nodes involved. In fact, some of them appear many times in different Biological Processes associated with the same interactome. All this suggests the important and central role of these genes in regulating some cellular functions related to COVID-19 [152], but it also raises questions we cannot yet answer today. For example, if the same gene appears in dozens of different Biological Processes, does this occur in a narrow window or over a long-time horizon? The analysis of cellular systems requires the coordination of large numbers of events, but identifying the temporal cues underlying interactions is the critical part of understanding cellular functions. With current knowledge, we could have a variety of interpretations, but they may be distorted [153]. This has led us to investigate the overall behavior of Biological Processes, rather than wanting to find the gold process at all costs.

Existing multiple interactions within the interactome show a complex network of gene regulation, in which some genes can influence a myriad of Biological Processes. However, when we say many genes and a “myriad of biological processes”, we need to know what we are talking about in quantitative terms. To my knowledge, no study related to SARS-CoV-2 has ever made such an assessment. To understand the similarity and dissimilarity of functions and the genes that support them, I used an analysis borrowed from marketing methods to compare the two data sets represented by the Biological Processes (GO). I compared the two interactomes through Data Merging (details in Methods, Section 2), combining the two large biological data sets into one (see Appendix A). Data Merging is used to evaluate interaction parameters, append observations, and find repetitions. Therefore, the logic I used was that to distinguish the common processes (coupled processes) from single processes (uncoupled processes) of each interactome. Merging the data optimizes the collection of all information into a single set, maximizing the completeness with which critical information can be extracted and analyzed. Appendix A also reports in full all the genes involved in the single terms, both paired and unpaired for 68,300 genes, which are also reported (Appendix A). These genes are redundant, because the same gene can take part in dozens of different molecular processes, as shown below in Table 7. This table illustrates the general picture that emerges from the merging of the two data sets, both containing common processes, but also specific to one or the other data set.

Table 7 shows how the Data Merging reveals thousands of genes with widespread gene redundancy, but also many uncoupled processes. These results show the activities exerted by the S1 subunit alone in its one-to-one relationships (in 814) have a relevant functional incidence (53%). However, the large number of high-scored genes in the same processes also means that multiple genes will have to appear multiple times in Biological Processes associated with the same interactome. An average value of over twenty genes per process shows how difficult it is to single out a single signaling pathway, or even a metabolic process, and assign genes to it.

The observed differences in gene composition suggest that gene expression and its involvement can vary depending on the specific context, such as different tissue types, conditions, or stages of development. This can cause different genes to be highlighted, even at different times, within the same larger biological process. We should not overlook the different ways in which 68,003 genes can be organized into 2837 different processes. About twenty genes are actually responsible for many processes. The overall number of processes is 23^2837^, while S1 comprises 26^1515^. This is an astronomical number of combinations, which makes it clear why adequate and correct experimental data, and their control, are necessary to reduce the combinations to a few when studying specific functional processes in any design context. As an illustration, when examining IL12A, involved in coupled processes, or RACK1, involved in uncoupled processes of 814 (Table 7), they exert a wide range of biological functions, so many that each of them is involved in over 100 processes. Therefore, how can we ascribe the precise biological pathway in which each of these proteins takes part, considering their abundance of over 100 occurrences within the interactomes under investigation?

Studies on HeLa cells have revealed that protein expression levels exceeding 90% are consistent with the average level of protein expression [154]. This shows that there is ample evidence to support an excess of protein copies, even at the level of gene expression, encompassing a significant portion of transcripts that encode functional proteins. But this ensures the efficient functioning of the processes in which these proteins are involved [154,155,156]. Protein abundance can be determined by many factors, such as transcription, translation, or RNA/protein decay [157]. Therefore, these factors can combine to produce a certain expression value. The load balance between transcription and translation regulates the gene expression necessary to optimize cellular fitness [158]. Low expression of essential proteins slows growth [159], but even generalized overexpression of proteins slows growth because it increases metabolic load [160] and energetic costs. Today, we can only say that the implications of over-representations of genes in an interactome can be multiple and each hypothesis influences the understanding of disease and cellular interactions [161]. The correct regulation of genes in space is necessary for proper function.

These claims may raise many questions, but there is no clear evidence to support any hypothesis or claims made about this matter. Despite technological advances in high-throughput sequencing, our ability to draw functional conclusions from expression data is lagging and qualitative [162,163,164]. The cell organizes its biochemistry in space by forming distinct chemical compartments in which membranes are separating barriers. Achieving the ability to differentiate the functions of cells within a multicellular tissue requires standardizing spatial transcriptomics data and correlating it with cellular mappings using bioinformatics systems. This will enable identifying various subpopulations with their distinct transcriptional profiles [165,166]. In addition, when we evaluate protein–protein interactions present in an interactome, we realize that, despite the integrations between different sources, they are far from complete in experimental terms [167,168]. This can lead to gaps in the real physical characterization and certainty of the interaction that is reflected in distortions of functional knowledge in GO processes. Superimposition between gene sets can cause low specificity in over-representation analysis, affecting the results and conclusions. Thus, over-representation (also called enrichment analysis) in genomic analysis plays a crucial role in several aspects. It works by identifying pathways or gene/protein sets that have a higher overlap with a known gene/protein set of functional interest than expected by chance. For example, it helps identify significant biological pathways associated with certain conditions or diseases by revealing how over-represented genes/proteins interconnect. The interconnectivity of genes, i.e., their membership in functional communities, enables us to unravel complex biological mechanisms that we cannot resolve by analyzing some individual processes or signaling pathways. In summary, over-representation is fundamental to interpreting genomic data, but when these are overabundant and complex, with high protein redundancies, as we find them here, it may be more appropriate to identify sets of genes that are interconnected and that exert specific functional activities in common. This way, we should have a more precise vision of the functional strategies in an interactome. Therefore, I eliminated redundancies from the three gene sets by isolating the single copy of each coding gene. I obtained three sets of coding genes: 944 genes for the coupled processes of interactomes 1060 + 814, 689 for the uncoupled processes of interactome-1060, and 771 for the uncoupled processes of interactome-814. I performed a clustering analysis of each of the three sets of their decoded products (Appendix A). The sets encompass proteins related to common and interconnected functional processes (1060 + 814), proteins involved in the one-to-one activity of S1 (814), and proteins derived from interactome-1060 that do not fall into the sets.

### 3.8. Clustering Analysis

I conducted this analysis on the three sets of coding genes to obtain an overall picture of the activities exerted by each set. Appendix A also reports the three sets of genes involved. Table 8, Table 9 and Table 10 show the overall results.

Although I also reported data on uncoupled functions of interactome-1060 and those coupled via the merging protocol, my analysis currently focuses only on one-to-one interactions of S1, but I will discuss uncorrelated data later to provide a broader perspective.

The list of Biological Processes and pathways provided by the clustering analysis (Table 8) reflects a complex interplay of metabolic activities influenced by the one-to-one interaction of the S1 subunit of the SARS-CoV-2 Spike protein. Many of these activities are central to the body’s response to infection, immune regulation, and cell signaling, and can be disrupted during both viral infection and vaccination. No one can rule this out. The clustering results cover broad macroscopic areas of activity: 1. Immune system activation and regulation; 2. Vascular and cardiovascular implications; 3. Metabolic processes; 4. Cell signaling and structural integrity; and 5. Neural and cognitive processes.

#### 3.8.1. The Liver’s Characteristics

The emergence of the hepatitis B pathway (hsa055161) is unexpected in SARS-CoV-2. The liver is one of the organs most affected by COVID-19, and an increase in liver enzymes is the most common symptom [169]. There appears to be a correlation between the severity of the disease and older patients with other morbidities. Chronic HBV infection can lead to metabolic syndrome and liver dysfunction [170]. The pathways involved in liver metabolism may intersect with systemic responses to SARS-CoV-2, especially in patients with pre-existing liver conditions. Metabolic dysfunction (MASH) is common in Western countries and proceeds through a slow progression of inflammation and fibrosis, which is associated with an imbalance of lipid metabolism and insulin resistance, components also common to COVID-19. This can exacerbate liver effects in chronic patients. The virus infects cholangiocytes with elevated levels of IL1, TNFA, and MCP1, all potential factors that can induce the development of MASH with progression to advanced chronic states, but we cannot exclude possible cancerous states [171]. It appears the hepatitis B path may come from shared immune mechanisms or pathways between SARS-CoV-2 and hepatitis B virus (HBV), relating to immune evasion strategies or overlapping receptor usage.

I could consider potential explanations: (A) Cross-reactivity: the immune response triggered by the S1 subunit may have shared components or epitopes with the hepatitis B virus, resulting in a cross-reactive immune response that affects hepatitis B-related pathways as well [172]. (B) It is conceivable that this could be a statistical anomaly or data noise, suggesting an indirect association not caused by the S1 subunit, but reflecting shared cellular machinery or immune pathways. C) Certain signaling pathways that are activated during viral infections, such as mTOR or immune-related pathways, play a role in the response to different viral infections, including hepatitis B, resulting in concurrent pathway activation [173,174]. Both viruses induce strong inflammatory responses, resulting in the activation of similar pathways in host cells and starting shared Biological Processes. The activation of signaling pathways, such as the mTOR pathway, in response to viral infection, serves as an illustrative example of a common theme. D) The metabolic alterations induced in host cells by both viruses facilitate the promotion of viral replication. As an example, HBV modifies lipid metabolism, a potential pathway that SARS-CoV-2 also affects via its protein. The overlapping Biological Processes highlight the intricate interplay between viral infections, immune responses, and cell metabolism, even though the S1 subunit of SARS-CoV-2 and hepatitis B might not link to direct metabolic activities. Understanding these connections can help explain the broader implications of viral infections on host health and developing vaccines. Further research would be essential to clarify the specifics of these relationships. However, considering the results of the Data Merging analysis, all this seems to be the effect of the huge number of genes involved, and the possibility of innumerable interactions with the same groups of overlapping molecules, to which I must add the scarcity of experimental data, all factors that can mislead even advanced computing systems.

#### 3.8.2. Vascular Aspects

The renin–angiotensin system is crucial for blood pressure regulation and fluid balance [175], and its involvement may explain some of the cardiovascular manifestations seen in COVID-19. In chronic liver disease, alterations in this system can exacerbate portal hypertension and fluid retention. Calmodulin binding connects and regulates MTOR, the renin–angiotensin system, blood vessel diameter maintenance, and vascular smooth muscle connection through the calcium signaling path. The latter also acts both on FC gamma R-mediated phagocytosis and Cytoskeleton regulation, driven by Integrin- and Integrin-mediated cell adhesion [176]. Immune cells, such as macrophages, use calcium signaling [177] to engulf and eliminate infected cells, including those that would be affected by hepatitis B. Integrins mediate cell–cell and cell–ECM interactions, influencing cell migration and signaling [178]. In liver injuries, integrin signaling can affect hepatocyte survival and regeneration [178,179]. During infection or immune response, disruptions in lipid metabolism, such as glycerophospholipid metabolism (hsa00564) and cholesterol metabolism, may occur, causing alterations in lipid profiles and contributing to the hypercoagulable states observed in COVID-19.

#### 3.8.3. Cumulative Effects May Cause Cancer Involvement

Certainly, the cumulative effects of chronic inflammation, metabolic dysfunction, and abnormal signaling pathways can increase the risk of hepatocellular carcinoma in individuals with underlying liver disease. However, we should also consider that these cumulative effects may promote oncogenesis. In this context, the mTOR pathway is crucial in regulating cell growth, proliferation, and survival, but it is often associated with cancer [180]. Molecular changes, such as mutations in oncogenes and tumor suppressor genes, can further drive any cancer development. Even in the other two clustering analyses, we can see connections with possible cancer progression, especially in terms of genomic instability, increased proliferation, immune evasion, and metastasis. Dysregulated kinase, such as PI3K and ECM–receptor, may support cancer progression [181,182].

Ribosome biogenesis and chromatin assembly might also lead to uncontrolled cell growth [183,184]. Targeted projects are necessary in these areas to obtain concrete answers and reveal significant signals of cancerous evolutions. Here, I only show that cancer evolution could be possible because the specific processes are active and in common.

#### 3.8.4. Neural Effects

The neural and cognitive processes are another important point. LTD, “Long-term Depression”, and “Calmodulin Binding” are involved in neural signaling and plasticity [185], suggesting potential effects on neurological and/or cognitive functions, which aligns with reports of neurological complications observed in COVID-19 patients [186,187,188]. Acting on both “FC gamma R-mediated phagocytosis” and “Cytoskeleton regulation” [189], driven by “Integrin” and “Integrin mediated cell adhesion”, it can affect LTD [190]. I have found LTD (hsa04730, *p*-value: 7.21 × 10^−92^) connected to “LTP, Long-term Potentiation” (hsa04720, *p*-value: 2.16 × 10^−32^), with many genes in common (see Figure 12).

The brain’s actions involve the participation and connection of both molecular processes. But there is also a potential link between these neurological processes and the S1 protein that could provide some clues about the molecular basis of the neurological impact of COVID-19. PRKCG, MAPK1, BRAF, KRAS, and ITPR1 are part of key signaling pathways like the MAPK/ERK pathway, which is often linked to cellular stress responses, inflammation, and apoptosis. Various COVID-19-related pathologies, particularly those affecting the immune response and inflammation in different tissues, including the brain, implicate them, especially in the MAPK/ERK pathway, which is often linked to cellular stress responses, inflammation, and apoptosis [191,192]. NOS1 (nitric oxide synthase 1) is involved in producing nitric oxide, a molecule with widespread roles in neurotransmission and vasodilation. Researchers have implicated the dysregulation of nitric oxide in COVID-19 in relation to endothelial dysfunction, which can also affect the brain [193]. Genes of the phosphoprotein phosphatase family, like PPP2CA, PPP2CB, and PRKG1, are involved in signaling cascades related to protein phosphorylation that tune platelet aggregation [194], but their dysregulation could contribute to the virus’s ability to manipulate cellular environments [195]. In addition, these genes have connections to inflammation, oxidative stress, and synaptic plasticity, processes that are modified during viral infections and potentially contribute to the long-term neurological consequences of COVID-19. PRKCG (Protein kinase C gamma) and MAPK1 have also been shown to modulate insulin signaling and glucose uptake [196] in the brain [197]. Disruptions in insulin and glucose metabolism pathways could contribute to neurological symptoms, including brain fog and fatigue reported in long COVID-19, as these pathways are tied to cognitive function. However, a very pertinent observation is the altered glucose metabolism in the brain reported by two French research groups [198,199]. In terms of glucose metabolism, genes like RAF1 and MAPK1 regulate metabolic homeostasis, including effects on glucose uptake and insulin sensitivity.

Genes like GNAI2 and GNAI3 (G-protein subunits) are part of G-protein-coupled receptor (GPCR) signaling pathways, which are involved in neurotransmitter systems, including serotonin signaling [200]. Serotonin regulation in the brain is crucial for mood, cognition, and overall neurological function. Dysregulation in this pathway could contribute to both the mood disorders and cognitive symptoms seen in long COVID-19 patients.

To my knowledge, these results show the first molecular evidence that COVID-19 may affect brain metabolism, because of these genes’ involvement in critical brain functions, synaptic plasticity, and metabolic pathways. They potentially contribute to neuroinflammation states and energy dysregulation, affecting cognitive performance. However, further experimental and computational work should merge these links to reveal new therapeutic targets.

## 4. Discussion

A multitude of studies have clarified the fundamental processes related to the S1 subunit of the Spike protein of SARS-CoV-2. The protein has garnered significant attention in scientific research because of its pivotal role in viral entry into host cells and its potential implications for immunogenicity and pathogenesis [201,202,203,204,205]. In a previous investigation of the liver during the COVID-19 pandemic, I observed the interaction of S1 with specific human proteins, ACE2, AGER, ESR1, FKBP4, KIF18A, MED1, NEK7, PRC1, RRAGC, S100A8, SFN, TLN1, TLR4, and TMPRSS2. I considered it an interesting anomaly [10], without delving into the matter. Here, I found only three proteins that coincide with those identified in that study (ACE2, ESR1, and TLR4). This seems to show a disparity in the respective metabolic contexts.

The S1 protein, through its one-to-one interactions, has opened a window into the metabolic strategies of SARS-CoV-2. Beyond the specific and solitary actions of S1 that could also be real for the vaccine, the general picture I observed has revealed many surprises. Gene redundancy within the interactome suggests the existence of a complex gene regulatory network, in which some genes can influence metabolic processes through a complex network of internal and external signals. The first consideration is concerning because of the enormous number of genes and proteins that operate in the cell involved in specific functions related to the disease. What perhaps we do not consider enough is that when we focus on a single functional process and try to attribute its constituent components to it, we do not consider that we must make choices among many combinations of components. Western blotting is not enough to say that if there are proteins, we also have the hypothesized process. The proteins are there, but these same proteins can be part of many processes. Only a canonical experimental approach gives us the certainty of what we are hypothesizing. The second consideration concerns the evolution that COVID-19 can have. There are some signals that suggest thrombophilic pathologies both following the disease and vaccination, but they are the unexpected signals that should make us reflect. Finding the statistical possibility of progression to hepatitis B among the results is puzzling. Most likely, it is a statistical consequence of the overlap of many similar processes in the two viruses. However, it prompted us to look at the results from a different angle.

### 4.1. Considerations on Cancer Development

The results of the interactomic analysis of the SARS-CoV-2 S1 Spike subunit reveal a fascinating and complex network of Biological Processes, some of which appear to be associated with cancer development. Let us analyze them and discuss the potential implications. S1 carries out its peculiar solitary activity when inside the infected cell, where it also operates together with other viral proteins, attacking individual human proteins with multiple interactions. In this context, mTOR dysregulation is associated with various cancers [206], making this a significant finding. The strong association also suggests that SARS-CoV-2 could affect pathways involved in cell growth and metabolism, potentially leading to oncogenic processes. PI3K is another critical pathway dysregulated in cancer [207]. Its presence in the results shows a potential link between SARS-CoV-2 infection and the activation of oncogenic signaling pathways. Nor can I neglect focal adhesion. This process is involved in cell adhesion, motility, and cell survival. Aberrant focal adhesion signaling plays a role in cancer metastases [208]. This GO term having a significant *p*-value shows a role in altering cellular environments that could predispose it to tumorigenesis.

### 4.2. Other Observations That Support Cancer Development

Other observations also support the same idea. Dysregulation of DNA replication and chromatin assembly can lead to genomic instability, a hallmark of cancer. The significant presence of these GO terms shows that SARS-CoV-2 could affect the fidelity of DNA replication, leading to mutations and cancer development. We often implicate cholesterol metabolism in cancer progression, particularly in lipid rafts. Lipid rafts are cholesterol-rich micro-domains that facilitate cell signaling, including pathways involved in cancer [209]. Chronic inflammation is also a well-known risk factor for cancer. The powerful signals in complement activation and coagulation pathways could suggest that SARS-CoV-2 could contribute to a pro-inflammatory and pro-coagulant state, which over time could lead to oncogenesis [210]. All this without considering the metabolic alterations induced by S1 in common with liver cancer that I have already discussed. My interactomic analysis suggests that S1, in SARS-CoV-2 infection, might contribute to cancer development through multiple mechanisms. These include many dysregulated mechanisms and liver-related complications as shown by the associating hepatitis B. While these findings do not establish a causal link between COVID-19 and cancer, they highlight potential areas for further research to understand how SARS-CoV-2 could contribute to cancer risk, especially in long-term survivors of infection. The vital role and multifaceted nature of these pathways cause continued exploration through thorough experimental studies. Therefore, I focused on TP53 and RPS27A, two peculiar high-ranking proteins of which interactions are present in the results. Their key features are that these proteins are involved in various ways in the viral/tumor progression of cells. To isolate and characterize their “world” in the interactome, I used the STRING action “recenter” that rewires the network on these proteins, showing all the proteins in STRING that interact with them. In Appendix A, I show the interactome. The results highlight an intricate network of protein interactions, centered on TP53, a crucial tumor suppressor protein, and RPS27A, a component of the ubiquitin–proteasome pathway. I analyzed the implications of these interactions in SARS-CoV-2 infection.

### 4.3. TP53 Interactions

TP53 is involved in maintaining genomic stability, cell cycle arrest, and apoptosis, but its role may vary depending on its binding partners [10]. In viral infections, manipulations of TP53 can favor either cellular defense or viral replication [211]. Interaction with proteins such as ATR, CHEK1,2, BRCA1, and DDB2 signals the activation of DNA repair pathways and can lead to cell cycle arrest [212]. These mechanisms would favor the cell by preserving genomic integrity and preventing proliferating virus-infected cells. In addition, interactions with BAX, BAK1, and CASP8 suggest the activation of apoptotic pathways. This is part of the cell defense mechanisms to eliminate virus-infected cells, especially before the virus can replicate. There is also transcriptional regulation through interactions with CREBBP, EP300, and SP1, which could stimulate expressing genes that protect the cell from viral attacks (pro-apoptotic genes or antiviral responses to interferon) [213]. The interaction with TIGAR (TP53 Induced Glycolysis Regulatory Phosphatase) and SLC2A1/SLC2A2 (glucose transporters) supports all of this at the metabolic level, potentially inhibiting viral replication. This occurs by limiting the glucose and directing the pathway into the pentose phosphate shunt, as viruses depend on the host metabolism [214].

Faced with these mechanisms favorable to cellular defense, we can also highlight mechanisms that are favorable to the virus. MDM2 and MDM4, which promote its degradation, negatively regulate TP53 [215]. We know that SARS-CoV-2 proteins hijack the MDM2-TP53 axis to suppress TP53-mediated apoptosis, favoring viral survival and replication [216]. Interactions with MAPK1, MAPK3, and MAPK9 suggest the modulation of cellular signaling pathways. SARS-CoV-2 can activate MAPKs to promote viral replication and evade immune responses [217]. We must also consider interactions with MAPK1, MAPK3, and MAPK9. Antiapoptotic mechanisms facilitate initial viral multiplication, while in the infection’s progression, apoptosis through cell lysis favors the release of virions.

Researchers liken the functional diversity of p53 to an incessant “tug of war” [218] because opposing functional processes create metabolic uncertainty.

#### RPS27A Interactions

RPS27A (a precursor of ubiquitin and ribosomal protein S27A) plays a key role in ubiquitination, which regulates protein degradation and signaling pathways. Its interaction with TP53, MDM2, and ubiquitin-related proteins such as UBE2D1/D2/D3, UBB, UBC, and USP7 suggests RPS27A is an integral part of controlling TP53 stability [219,220]. Here too, we can evaluate cell-protective actions and mechanisms that favor the virus.

Proper regulation of ubiquitin-mediated proteolysis (Ubiquitination and Proteasomal Degradation) is essential to remove damaged proteins and maintain cellular health. Interactions of RPS27A with proteasome-related components (e.g., CUL1, SKP1, RBX1) can regulate the degradation of viral proteins, preventing viral assembly and replication [221]. But SARS-CoV-2 exploits the host ubiquitin–proteasome system to degrade antiviral proteins and facilitate its own replication [222]. The interaction of RPS27A with MDM2, a key regulator of TP53 degradation, suggests that viral infection could lead to TP53 inactivation by promoting its degradation via ubiquitination.

In addition, interactions with deubiquitinating enzymes such as USP7 and UBE2I play a critical role in regulating TP53 and RPS27A [199]. Both USP7 and UBE2I are involved in the removal of ubiquitin moieties, which influences the stability and function of proteins, especially key regulators such as TP53. The virus can exploit this regulatory mechanism to weaken cellular defenses, influencing the stability and function of the two proteins. If viral manipulation distorts these pathways, it could compromise the cell’s ability to mount an effective defense. SARS-CoV-2, as with other viruses, hijacks the host ubiquitin–proteasome system to evade immune responses [221]. By interacting with USP7, UBE2I, and other ubiquitin-related enzymes, the virus could do the following: 1. Protect viral proteins from degradation. 2. Suppress TP53 activity by promoting its degradation [213,215,220] or reducing its pro-apoptotic function via SUMOylation [222]. 3. Modulate immune responses by preventing the activation of key antiviral pathways [217].

Recent papers [223,224,225,226] are providing insights into how Spike proteins of many variants behave in interactions, and are focusing on the latest advances in strategies for developing inhibitors of these interactions. They study the regulations of genes/proteins, highlighting their implications for the biology of the virus. Researchers have also studied some interactions using the sequences of the S1 subunit and bioinformatics algorithms to identify potential sites for interaction with human proteins.

## 5. Conclusions

This study highlights the multifaceted roles of the S1 subunit in immune modulation, metabolic reprogramming, and systemic effects. It underscores the importance of S1 in understanding COVID-19 pathophysiology. It confirms many of the observations already known on SARS-CoV-2 and COVID-19, to which it attributes a more organic role. In addition, it uncovers novel functional associations and shows the extensive repertoire of implicated genes, including a significant proportion involved in diverse processes.

The interactomic analysis reveals a complex network of interconnected Biological Processes, some of which are associated with cancer development and cognitive effects. S1 affects overall metabolism by altering energy production, influencing lipid metabolism, modulating immune responses, and affecting systemic inflammatory processes. These changes not only support viral replication but can also lead to various metabolic disturbances in the host, contributing to the overall pathology associated both with COVID-19 and vaccination. Understanding these mechanisms is crucial for developing effective treatments and vaccines.

The unexpected interplay between hepatitis B and COVID-19 involves pathophysiological mechanisms that may exacerbate previous or under-observed clinical liver pathology, complicating patient treatments. However, even if there might not be an explicit link between the direct metabolic activities related to S1 and hepatitis B, these overlapping Biological Processes highlight an intricate interplay between viral infections, immune responses, and cell metabolism. Understanding these connections is crucial to explain the precise mechanisms regarding the inexplicable presence of hepatitis B-related processes or the broader implications of viral infections on host health. Many overlapping processes are also common to cancer progression. Thus, we cannot exclude anything.

My results reveal a network of Biological Processes, some of which are indeed associated with cancer development. The interactomic analysis suggests that SARS-CoV-2 infection might contribute to cancer development through the dysregulation of key oncogenic pathways like mTOR and PI3K, disruption of DNA replication and chromatin assembly, proteolytic cleavage, chronic inflammation, and liver-related complications, as shown by the hepatitis B association. These activations, associated with proinflammatory mediator release, suggest an underlying activation of blood clotting-related gene expression by specific S1 interactions, which might predispose some individuals to inflammation-related anaphylaxis and blood clotting. Recent observations have highlighted the role of hematopoietic system aging in driving cancer progression through inflammation-induced impairment of immunity [227].

These findings do not establish a causal link between COVID-19 and cancer because of the complexity of these connections. However, they highlight potential areas for further research to understand how SARS-CoV-2 might contribute to cancer risk, especially in long-term survivors of the infection. For example, we could study the deubiquitinating enzymes and the same S1 as potential drug targets.

However, my results have some limitations. The wide-ranging assortment of genes derived from Data Merging required us to interpret the findings as collective properties that emerge from the causal topological structure induced by S1 and its functional dynamics. This justifies my resolution to present the results as sets of processes with the most statistically reliable functional characteristics, rather than going to find single functional processes. Another constraint is the vast number of overlapping processes occurring at the cellular level, as discussed, which rely on the data quality and quantity provided by interactomic mathematical models to researchers for their analyses.

Through applying my approach to the S1-induced interactome, I not only confirmed existing associations, but also unveiled unknown connections. I provided insights into the intricate modulation of gene expressions that underlie both normal and pathological functional processes. Cells with the same molecules can exhibit many and different phenotypic properties at multiple levels, making them difficult to define, classify, and understand. I integrated multiple approaches to have a coherent vision that, despite the known spatio-temporal limitations, gave us useful information. The cultural landscape in which research on deep cellular mechanisms falls is based on a static and timeless vision of the metabolism, which produces static data. The basis for the calculations of bioinformatics systems lies in these same data, leading to the formation of intricate and heterogeneous networks. However, interactomics is a mandatory intermediate step if we want to decode gene expression in its functional aspects. We should never forget that epigenetic changes, such as DNA methylation or histone modifications, influence gene expression and lead to many cellular responses, which manifest themselves in an innumerable number of different human phenotypes. Therefore, the result of any strategy, viral or cancerous, must first confront the phenotype of its host to progress.

## Figures and Tables

**Figure 1 biomolecules-14-01549-f001:**
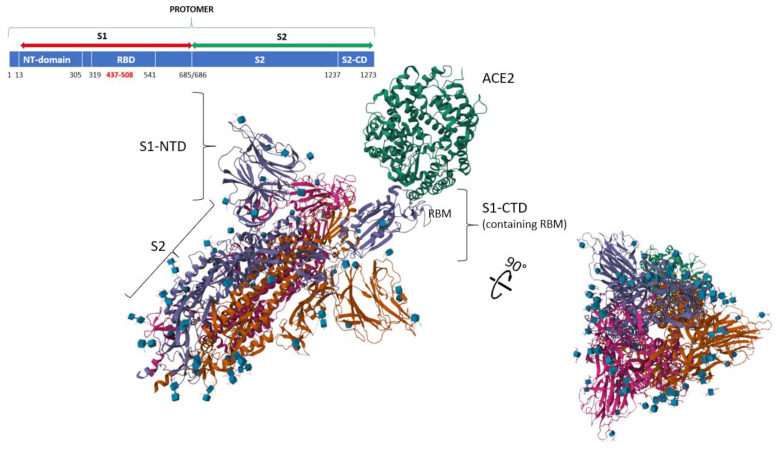
Model of the structural organization of the SARS-CoV-2 Spike protein complexed with ACE2 in a closed state [38]. PDB, the Expression System Homo sapiens, offers information about the trimeric complex, found at PDB-DOI: https://doi.org/10.2210/pdb7DF4/pdb. You can also access the structure at https://www.rcsb.org/structure/7df4 (accessed on 2 June 2024). I used PDB resources to process the images. Information in the article by Chao et al. [38] inspired the figure. The structure of ACE2 is on Uniprot, as Q9BYF1. The three identical protomers are in red, cyan, and brown. ACE2 is in green. The subunit that interacts with ACE2 is the one in cyan. Acronyms: S1-NTD, S1 N-terminal domain; S1-CTD, S1 C-terminal domain; RBM, Receptor-Binding Motif, interacting with ACE2; RBD, Receptor-Binding Domain. Some small cyan square formations surrounding the complex are molecules of NAG (2-acetamido-2-deoxy-beta-D-glucopyranose), a carbohydrate conserved in eukaryotic glycoproteins. The diagram at the top left sketches the structural organization of a Spike protomer, highlighting some relevant information. The 437–508 segment (in red) is the region that interacts with ACE2. S2-CD is the S2 C-terminal domain.

**Figure 2 biomolecules-14-01549-f002:**
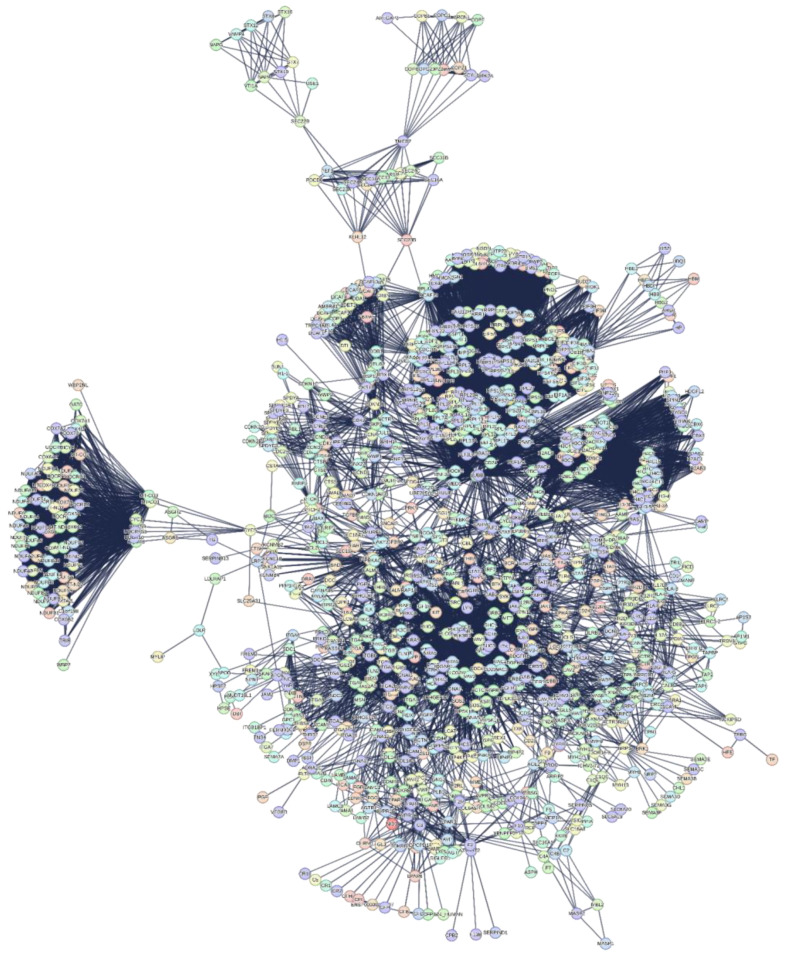
Human interactome generated by seed proteins physically interacting with S1 subunit of SARS-CoV-2. STRING calculated the interactome.

**Figure 3 biomolecules-14-01549-f003:**
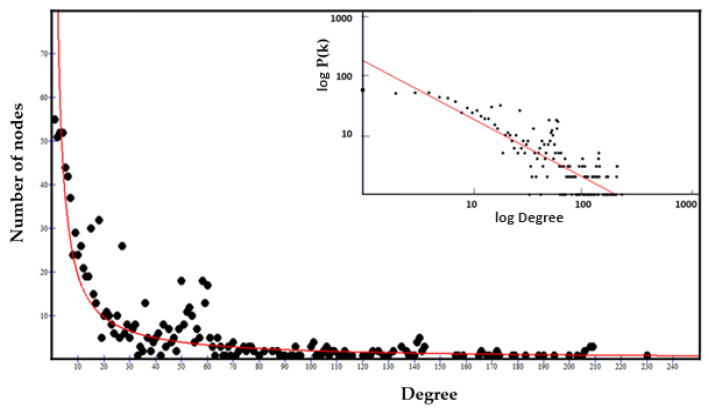
Power law distribution of interactome-1060. The data follows a scale-free distribution based on the power law. A log–log scale is shown in the inset. The best fit line in the log–log inset yields a slope of −0.517.

**Figure 4 biomolecules-14-01549-f004:**
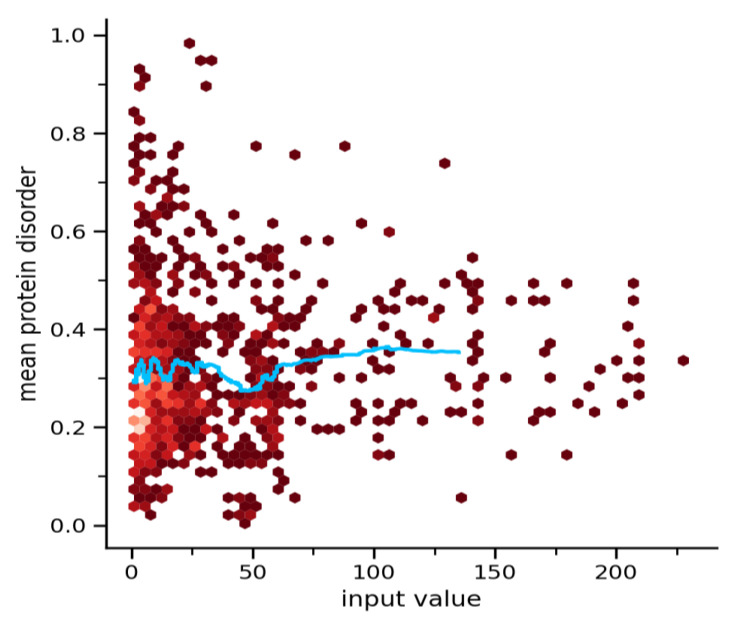
The plot shows the average distribution of the protein disorder of the associated proteins in interactome-1060. Pearson’s r value: −0.1; Pearson’s *p*-value: 0.0015; BP-R^2^: 0.062 (medium). The blue light line is the median. STRING computed disorder content from sequences. The measure of BP-R^2^ is based on checking how much the values of the specified trend property deviate from the mean. Its scale (from 0 to 1) follows a quadratic pattern and does not have a confidence measure associated with the BP-R^2^. As a result, STRING has included some thresholds and the value of 0.15 is medium.

**Figure 5 biomolecules-14-01549-f005:**
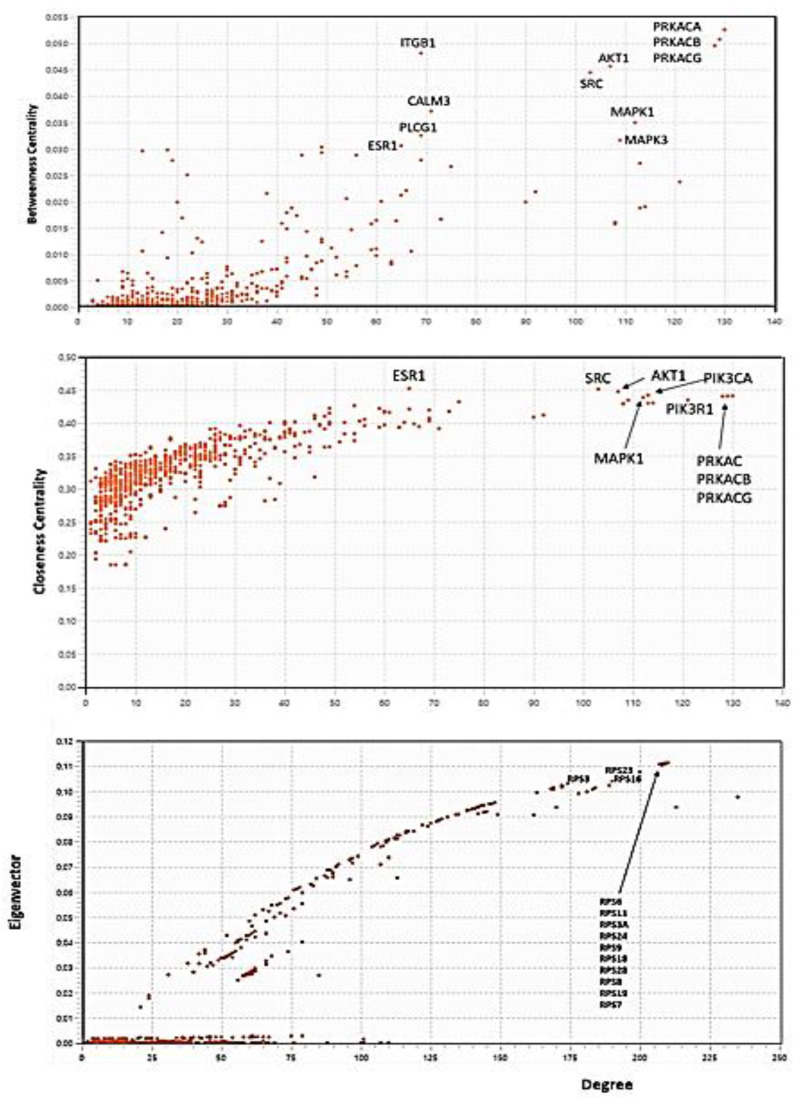
The centrality distribution plots of interactome-1060: closeness centrality (**top**), betweenness centrality (**middle**), and eigenvector (**bottom**). Using the Network Analyzer (version 4.5.0) in Cytoscape, the topological parameters were determined to find the node values [25,81,82]. The calculated distributions originate from the interactome.

**Figure 6 biomolecules-14-01549-f006:**
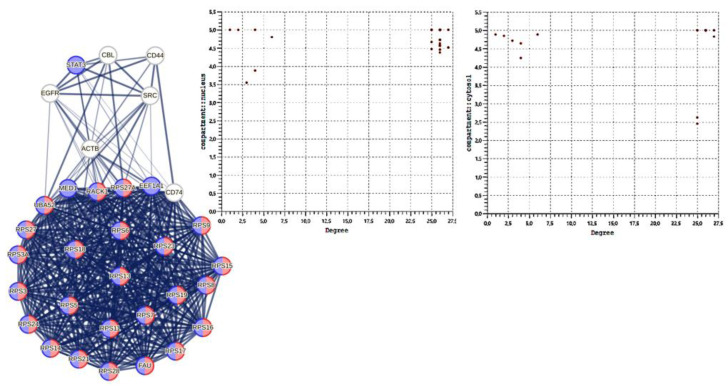
The graph shows the hub-spoke pattern (left) generated by the selected nodes. The red color shows proteins involved in cytoplasmic translations (GO:0002181; strength: 2.07 and *p*-value: 6.81 × 10^−41^), while the blue one shows proteins involved in gene expression (GO:0010467; strength: 0.89 and *p*-value: 4.00 × 10^−18^). STRING calculated the graph. Plots on the right show the distributions of nodes in the cellular compartments. These two classes of nodes operate in the cytosol and nucleus, some in both. Calculation performed by Cytoscape.

**Figure 7 biomolecules-14-01549-f007:**
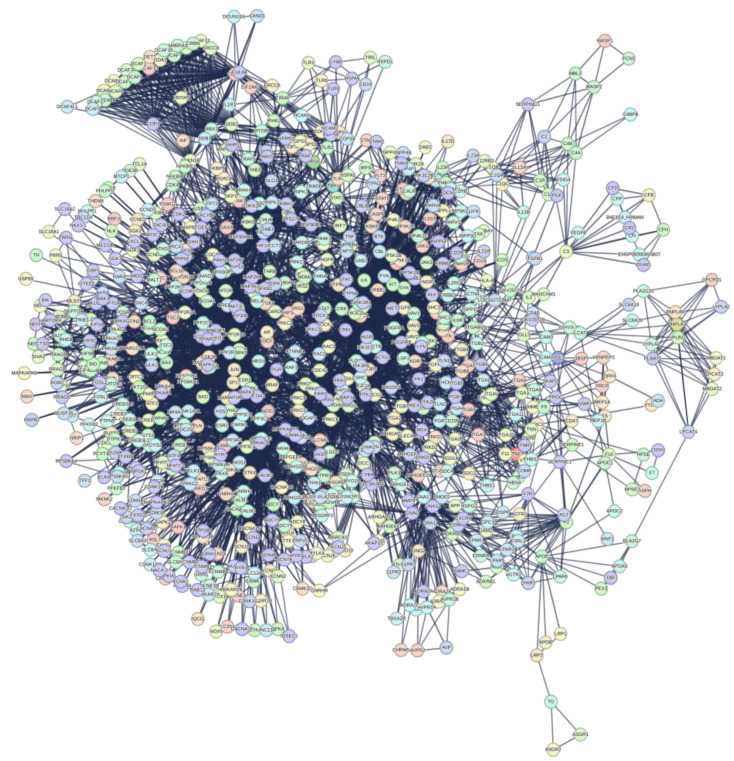
Interactome of the 27 human proteins interacting one-to-one with S1. Number of nodes: 814; number of edges: 7409; average node degree: 15.9; avg. local clustering coefficient: 0.547; expected number of edges: 2285; PPI enrichment *p*-value: <1.0 × 10^−16^; 6 channels (without Text Mining); confidence score of 0.900; enrichment: 500 1st order + 500 2nd order proteins.

**Figure 8 biomolecules-14-01549-f008:**
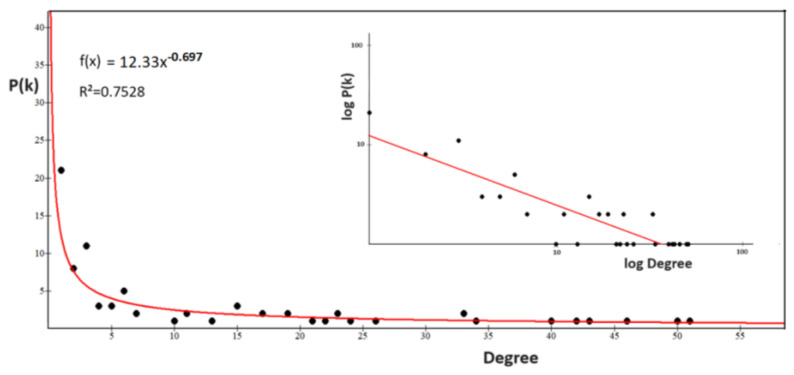
Power law distribution of interactome-814. The distribution follows a scale-free distribution based on the power law. In the inset, the same nodes are shown on a log–log scale with the best fit to the data (f(x) = 12.33 x^−.697^ and R^2^ = 0.7528). The slope is −0.374 and is calculated on the best fit line in the log–log inset.

**Figure 9 biomolecules-14-01549-f009:**
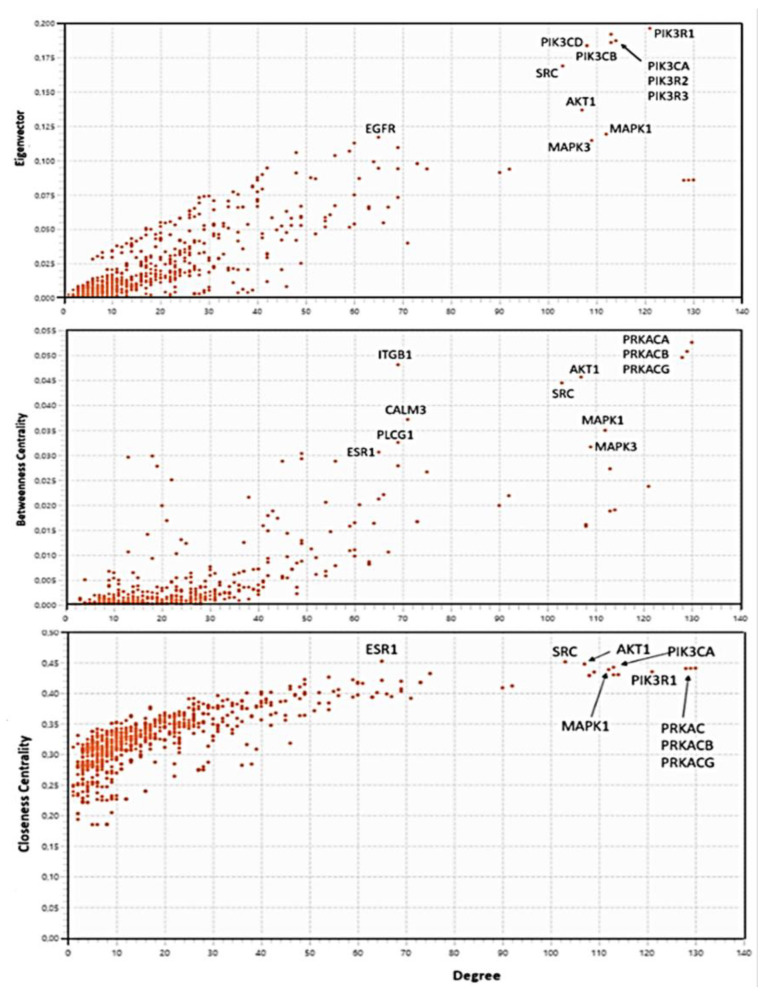
The centrality distribution plots of interactome-814: closeness centrality (**top**), betweenness centrality (**middle**) and eigenvector (**bottom**). I calculated the topological parameters using the Network Analyzer (version 4.5.0) on Cytoscape to identify the node values [82].

**Figure 10 biomolecules-14-01549-f010:**
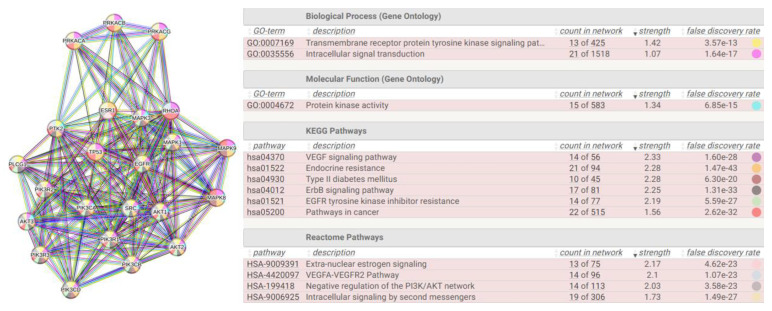
The hub-spoke organization of interactome-814. The lower central part of the graph (**left**) shows both pure hub proteins and many mixed ones. The table on the right shows some of the most significant biological terms regulated by these high-ranking nodes. It is interesting to note that each of the nodes can take part in multiple biological functions. The table was calculated by STRING.

**Figure 11 biomolecules-14-01549-f011:**
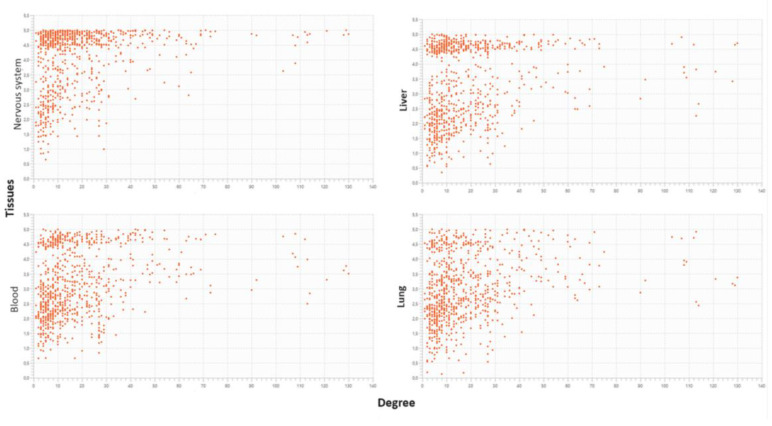
Distribution of interactome-814 proteins in cellular compartments (nucleus and cytosol) and tissues (blood and nervous system). Calculations performed by Cytoscape.

**Figure 12 biomolecules-14-01549-f012:**
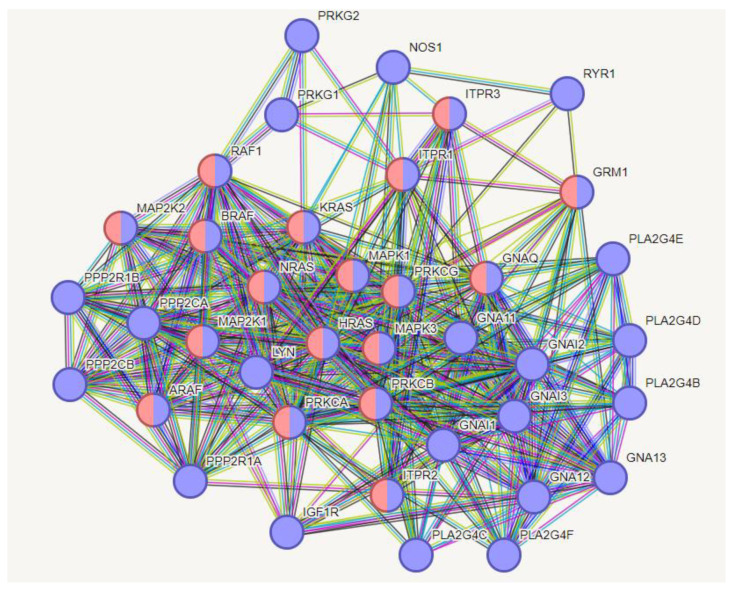
Relationships between LTP and LTD processes. These two molecular processes show many nodes in common. LTP (in red), Long-term potentiation (hsa04720), 18 nodes involved, strength 2,16 and *p*-value: 3.50 × 10^−32^. LTD (in blue), Long-term depression (hsa04730), 39 nodes involved, strength 2.52, and *p*-value: 7.21 × 10^−92^. LTD is a process involving a decrease in the synaptic strength with multiple signal transduction pathways involved. LTP is a long-lasting increase in synaptic efficacy. The high strength values show that many proteins support the involvement of multiple signal transduction pathways in both processes.

**Table 1 biomolecules-14-01549-t001:** Functional processes activated in the human genome by interactome-1060.

Biological Process	Terms Significantly Enriched
Biological Process (Gene Ontology)	1430 terms
Molecular Function (Gene Ontology)	165 terms
Cellular Component (Gene Ontology)	283 terms
**Reference publications (PubMed)**	**>10,000 publications**
Local network cluster (STRING)	251 clusters
KEGG Pathways	202 pathways
Reactome Pathways	693 pathways
WikiPathways	302 pathways
Disease-gene associations (DISEASES)	114 diseases
Tissue expression (TISSUES)	167 tissues
Subcellular localization (COMPARTMENTS)	287 compartments
Human Phenotype (Monarch)	787 phenotypes
Annotated Keywords (UniProt)	103 keywords
Protein Domains (Pfam)	17 domains
Protein Domains and Features (InterPro)	144 domains
Protein Domains (SMART)	44 domains
**All enriched terms (without PubMed)**	**4989 enriched terms**

Note: Terms in bold represent the scientific quality of the processes activated in the 1060 interactome (see also Appendix B for further explanation). It is interesting to compare them with those in Table 6 that concern the 814 interactome.

**Table 2 biomolecules-14-01549-t002:** Topological parameters of interactome-1060 *.

Number of nodes	1060
Number of edges	17493 **
Average node degree	33
Avg. local clustering coefficient	0.679
Expected number of edges	8382
PPI enrichment *p*-value	<1.0 × 10^−16^
Confidence score	0.900
Source channels	6
Network diameter	10
Network radius	5
Characteristic path length	3.717
Network heterogeneity	1.187
Network density	0.33
Network centralization	0.189
Connected components	1 ***

(*) Calculated by Cytoscape Network Analyzer, which computes a comprehensive set of topological parameters [81,82]. (**) The numerical value shown is half of that reported in the Appendix A, which refers to the total interactions present in the interactome (34,986). STRING in some of its calculations doubles the value because it considers the interaction of a pair (A-B) in the two directions (from A to B and from B to A). (***) The value of “1” shows that all nodes in the network are connected to each other. Existing unconnected components (0 ≤ C ≤ 1) alter the calculations of the topological parameters, making them unreliable [54]. This is the fundamental reason for pruning. A single component accounts for strong network community.

**Table 3 biomolecules-14-01549-t003:** High-ranked hub and bottleneck nodes of interactome-1060.

Hub Nodes	Degree	Bottleneck	Degree
RPS6	210	RPS27A	235
RPS11	209	UBA52	213
RPS3A	209	RACK1	149
RPS24	209	CD74	110
RPS9	209	MED1	107
RPS18	208	SRC	101
RPS28	209	EEF1A1	88
RPS8	208	EGFR	76
RPS19	207	ACTB	65
RPS7	208	CD44	48
RPS23	200	STAT3	49
RPS16	190	CBL	37
RPS3	174		
RPS15	172		
RPS5	189		
FAU	172		
RPS13	184		
RPS21	169		
RPS17	169		
RPS14	183		
RPS27	181		

**Table 4 biomolecules-14-01549-t004:** Topological parameters of interactome-814 *.

Number of nodes	814
Number of edges	7409
Average node degree	15.9
Avg. local clustering coefficient	0.547
Expected number of edges	2285
PPI enrichment *p*-value	<1.0 × 10^−16^
Confidence score	0.900
Source channels	6
Network diameter	7
Network radius	4
Characteristic path length	3.189
Network heterogeneity	1.042
Network density	0.22
Network centralization	0.138
Connected components	1 **

(*) Calculated by Cytoscape Network Analyzer, which computes a comprehensive set of topological parameters [81,82]. (**) This value is “1” to show that all nodes in the network are connected to each other. Existing unconnected components (0 ≤ C ≤ 1) alter the calculations of the topological parameters, making them unreliable [54]. This is the fundamental reason for pruning. A single component accounts for strong network community.

**Table 5 biomolecules-14-01549-t005:** High-ranked hub and bottleneck nodes of interactome-814.

HUB Nodes	Degree	Bottleneck Nodes	Degree
**PIK3R1**	121	**AKT1**	65
**PIK3CA**	113	**EGFR**	103
PIK3R2	114	**ESR1**	107
PIK3R3	113	**MAPK1**	113
PIK3CD	108	**MAPK3**	130
PIK3CB	108	**PIK3CA**	129
**SRC**	103	**PIK3R1**	128
**AKT1**	107	PRKACA	112
**MAPK1**	112	PRKACB	121
**EGFR**	65	PRKACG	109
**MAPK3**	109	PTK2	75
AKT3	73	RHOA	49
AKT2	73	**SRC**	65
**ESR1**	65	**TP53**	69
PLCG1	69		
**TP53**	75		
MAPK8	92		
MAPK9	90		

Note: the proteins in **bold** are both hub and bottleneck.

**Table 6 biomolecules-14-01549-t006:** Functional processes activated in the human genome by interactome-814.

Biological Process	Terms Significantly Enriched
Biological Process (Gene Ontology)	2557 terms
Molecular Function (Gene Ontology)	321 terms
Cellular Component (Gene Ontology)	231 terms
**Reference publications (PubMed)**	**>10,000 publications**
Local network cluster (STRING)	246 clusters
KEGG Pathways	213 pathways
Reactome Pathways	828 pathways
WikiPathways	453 pathways
Disease-gene associations (DISEASES)	222 diseases
Tissue expression (TISSUES)	223 tissues
Subcellular localization (COMPARTMENTS)	218 compartments
Human Phenotype (Monarch)	1196 phenotypes
Annotated Keywords (UniProt)	124 keywords
Protein Domains (Pfam)	14 domains
Protein Domains and Features (InterPro)	222 domains
Protein Domains (SMART)	52 domains
**All enriched terms (without PubMed)**	**7120 enriched terms**

Note: It is interesting to compare the bold terms with those in Table 1. They show how the interactome-814, although with a 23% lower number of nodes, shows a 30% higher number of functional activities.

**Table 7 biomolecules-14-01549-t007:** Data Merging between Biological Processes (GO) of interactomes 1060 and 814.

	Number of Biological Processes (GO) (%)	Redundant Genes (%) *	Coding Genes	Average Genes Per Single Process	Genes Found > 100 Times
Merging of 1060 + 814 (after pruning) **	2837 (total)	68,003 (total)	---	23.97	-----
Coupled processes in the merging of 1060 + 814	554 (39) ***	24,301 (35.8)	944	21.9	ABL1, AGT, AKT1, APOE, BCL2, BTK, CD28, EGFR, FYN, HLA, HRAS, IL12A, IL12B, IL12RB1, IL23A, JAK2, KDR, KIT, LYN, MAPK1, RHOA, SRC, SYK, THBS1, TICAM1, TLR4, TNF, TYK2, ZAP70.
Uncoupled processes in 814	1515 (53)	39,691 (58.3)	771	26.19	ADA, ADCY8, ADRA1A, ADRA2A, AGT, AGTR2, AKT1, AKT2, APOE, APP, AR, ASPH, ATF2, ATF4, ATP2B4, AVP, AVPR, BAD, BAK1, BAX, BCL2, CALM1, CTNNB1, CYBA, DLG1, EDNRA, EGFR, EP300, FOS, FOXO1, FOXO3, FYN, GNAI2, GSK3A, GSK3B, HIF1A, HSP90AA1, HSP90AB1, HSPA5, IGF1R, IL12B, IL2, INSR, IRAK1, ITGB1, JAK2, JUN, KCNE1, KCNQ1, KDR, KIT, LYN, MALT1, MAP2K1, MAPK1, MAPK14, MAPK3, MAPK8, MED1, MMP9, MTOR, MYD88, NFKB1, NKX3-1, NOS1, PODGFRA, PIK3CA, PIK3CG, PLCG2, PPARA, PPARG, PPP3CA, PRKCD, PTEN, PTK2B, PTPN2, RELA, RHOA, RIPK1, RIPK2, RACK1, RPTOR, SLC8A1, SMAD3, SNCA, SRC, STAT3, SYK, TGFB1, THBS1, TIRAP, TLR2, TLR4, TNF, TP53
Uncoupled processes in 1060	214 (8)	4011 (5.9)	701	18.74	Family EIF, Eukaryotic initiation factors gene family, (230), histones (295), family NDUF (352), family RPL (516), family RPS (411). ****

(*) Multiple representations are possible for each gene. They are redundant because they belong to multiple processes. (**) I merged 1060 + 813 by considering only those Biological Processes (GO) with a strength value >0.05 (see details in Methods, Section 2). (***) After merging, I found 554 similar coupled processes (compared one to one); thus, in absolute value, they correspond to 1108 single processes. (****) in parentheses, the number of genes that make up the family.

**Table 8 biomolecules-14-01549-t008:** Clustering analysis of coding genes from Data Merging: Clusters of Uncoupled Functions of Interactome-1060.

Cluster No.	Primary Description	GO-Term	*p*-Value	Gene Count *
1	Cytoplasmic translation	GO:0002181	4.83 × 10^−83^	266
2	Focal adhesion	GO:0005925	7.61 × 10^−48^	189
3	Aerobic electron transport chain	GO:0019646	1.49 × 10^−47^	75
4	DNA replication-dependent chromatin assembly	GO:0006335	6.67 × 10^−19^	44
5	Antigen processing and presentation	GO:0019882	6.67 × 10^−16^	33
6	Complement activation, classical pathway	GO:0006958	1.67 × 10^−11^	23
7	COPII vesicle coat	GO:0030127	2.46 × 10^−12^	20
8	Activation of phospholipase C activity	GO:0007202	3.30 × 10^−6^	18
9	COPI vesicle coat	GO:0030126	1.90 × 10^−9^	11
10	Cholesterol metabolism	hsa04979	2.70 × 10^−4^	10
**Cluster No.**	**Secondary description**	**GO-term**	***p*-value**	**Gene count**
1	Formation of a pool of free 40S subunits	HAS-72689	7.09 × 10^−91^	-
3	Respiratory chain complex	GO:0098803	7.29 × 10^−52^	-
4	CENP-A containing nucleosome	GO:0043505	5.51 × 10^−15^	-
6	Complement and coagulation cascades	hsa04610	4.06 × 10^−9^	-
8	G alpha (q) signaling events	HAS-418597	1.11 × 10^−3^	-
10	Plasma lipoprotein particle clearance	GO:0034381	5.60 × 10^−3^	-
**Cluster No.**	**Tertiary description**	**GO-term**	***p*-value**	**Gene count**
1	Ribosome	GO:0005848	2.08 × 10^−79^	-

(*) The gene counts are total for the three descriptions of each cluster, so I report them only next to the primary description. The cluster numbers reported in the first column on the left for the secondary and/or tertiary descriptions are associated with those of the primary description. For example, the association of (1) Cytoplasmic translation + (2) Formation of a pool of free 40S subunits + (3) Ribosome, involving 266 genes, forms Cluster No. 1 of interactome-1060.

**Table 9 biomolecules-14-01549-t009:** Clustering analysis of coding genes from Data Merging: Clusters of Coupled Functions of Interactomes-1060 + 814.

Cluster No.	Primary Description	GO-Term	*p*-Value	Gene Count
1	Positive regulation of transferase activity	GO:0051347	2.76 × 10^−63^	409
2	Focal adhesion	GO:0005925	5.66 × 10^−44^	232
3	ECM–receptor interaction	hsa04512	9.88 × 10^−36^	79
4	Long-term potentiation	HAS-9620244	7.01 × 10^−6^	54
5	Rho protein signal transduction	GO:00072666	9.12 × 10^−8^	43
6	Formation of Fibrin Clot (Clotting Cascade)	CL:18784	1.09 × 10^−6^	37
7	Antigen processing and presentation	GO:0019882	7.05 × 10^−13^	35
8	Complement activation	GO:006956	1.33 × 10^−18^	33
9	Cholesterol metabolism	hsa04979	1.80 × 10^−3^	13
10	Renin–angiotensin system	hsa4614	2.09 × 10^−3^	9
**Cluster No.**	**Secondary description**	**GO-term**	***p*-value**	**Gene count**
1	Cellular responses to stress		7.56 × 10^−11^	-
2	Mixed, incl. Constitutive Signaling by Aberrant PI3K in Cancer, and FCERI mediated Ca + 2 mobilization	CL:17328	2.28 × 10^−34^	-
3	Protein complex involved in cell adhesion	GOCC:0098636	1.09 × 10^−27^	-
4	Calmodulin binding	KW.0112	5.05 × 10^−15^	-
5	G alpha (12/13) signaling events	HAS-416482	1.69× 10^−9^	-
6	Blood coagulation	GO:0007596	7.29 × 10^−24^	-
9	Regulation of plasma lipoprotein particle levels	GO:0097006	2.16 × 10^−5^	-
**Cluster No.**	**Tertiary description**	**GO-term**	***p*-value**	**Gene count**
1	Protein kinase binding	GO:0019901	6.94 × 10^−74^	-
5	Mixed, incl. Sema4D in semaphorin signaling, and ARHGEF1-like, PH domain.	CL:17973	1.765× 10^−6^	-
9	Protein–lipid complex	GO:0032994	6946 × 10^−74^	-

**Table 10 biomolecules-14-01549-t010:** Clustering analysis of coding genes from Data Merging: Clusters of Uncoupled Functions of Interactome-814.

Cluster No.	Primary Description	GO-Term	*p*-Value	Gene Count
1	Hepatitis B	hsa055161	4.98 × 10^−73^	259
2	mTOR signaling pathway	hsa04150	2.05 × 10^−36^	139
3	Fc gamma R-mediated phagocytosis	hsa04555	6.62 × 10^−32^	113
4	Long-term depression	hsa04730	1.72 × 10^−29^	72
5	Blood vessels diameter maintenance	GO:0097746	3.61 × 10^−13^	61
6	ECM–receptor interaction	hsa04512	9.96 × 10^−24^	56
7	Complement activation	GO:0006956	3.73 × 10^−18^	32
8	Renin–angiotensin system	hsa04614	1.10 × 10^−4^	14
9	Glycerophospholipid metabolism	Hsa00564	2.71 × 10^−8^	13
10	Plasma lipoprotein particle remodeling	GO:0034369	9.94 × 10^−5^	12
**Cluster No.**	**Secondary description**	**GO-term**	***p*-value**	**Gene count**
3	Constitutive Signaling by Aberrant PI3K in Cancer		1.12 × 10^−33^	-
4	Calmodulin binding	GO:0005516	1.11 × 10^−21^	-
5	Mixed, incl. Heterotrimeric G-protein complex, and Signaling transduction inhibitor	CL24307	6.90 × 10^−13^	-
6	Cell adhesion mediated by integrin	GO:0033627	2.32 × 10^−14^	-
7	Initial triggering of complement	HAS-166663	5.26 × 10^−12^	-
8	Dipeptidyl-peptidase activity, and Meprin A complex	CL31769	6.08 × 10^−3^	-
10	Cholesterol metabolism	hsa04979	1.98 × 10^−49^	
**Cluster No.**	**Tertiary description**	**GO-term**	***p*-value**	**Gene count**
3	GPVI-mediated activation cascade, and SH2 domain superfamily	CL:17470	1.53 × 10^−27^	-
5	Vascular smooth muscle contraction	hsa04270	8.57 × 10^−39^	-
6	Integrin	KW-0401	1.88 × 10^−16^	-
10	Protein–lipid complex	GO:0032994	5.40 × 10^−3^	-

## Data Availability

Data supporting the reported results are available in Appendix A, which were generated during this study, as mentioned in the Appendix A.

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
