# Peer review of "Interactomic Analyses and a Reverse Engineering Study Identify Specific Functional Activities of One-to-One Interactions of the S1 Subunit of the SARS-CoV-2 Spike Protein with the Human Proteome"

_biomolecules, 2024, doi:10.3390/biom14121549_

Round 1

Reviewer 1 Report

Comments and Suggestions for Authors

The manuscript provides comprehensive analyses of 2 SARS-CoV-2 S1 subunit interactomes, referred to as interactome-1060 and interactome-814, aimed at understanding their biological implications and drawing comparisons with hepatitis B virus and cancer development. While the study is valuable and aligns well with the journal's scope, I recommend a major revision for the following reasons:

1.        Length: The manuscript is excessively long. I feel it could benefit from streamlining to improve clarity and conciseness. While the analysis is thorough and comprehensive, some technical sections can be condensed or moved to supplementary materials to make the manuscript focus on the biological implications.

2.        Abstract: The abstract would benefit from a brief introductory sentence to provide context on the SARS-CoV-2 viruses and S1 subunit, allowing readers to better understand the context.

3.        Introduction: 1) The introduction should be more concise. For example, the first two paragraphs can be combined to streamline the background. Also, the paragraphs explaining why S1 is being studied, the use of PPI for studying S1, and the impact of post-translational modifications on the interatomic analysis (lines 57-132) should be shortened for clarity. The effect of post-translational modifications is discussed later in the manuscript, this part can be removed from introduction to avoid repetition. 2) Include an explanation of ACE2 to help readers better understand.

4.        Results: 1) Consider moving Figure 1S, which shows the structure of S1 unit, to the main text (section 3.1) to help readers better understand and visualize the S1 subunit. Also, the figure would benefit from a more professional appearance. Instead of a screenshot from PyMOL, create the diagram with a white background and clearly labeled bars (see the attached figures for more information). 2) Line 375 references "pdbj"; it should be corrected to "PDB" to avoid confusion. 3) Streamline the discussion of network centralities and topological parameters or move these details to supplementary materials. 4) In lines 1337-1351, some terms are unnecessarily placed in quotation marks; this should be clarified or revised.

5.        Figure Presentation: The presentation of figures needs improvement. 1) In the protein interactome maps, the nodes should be analyzed and categorized to help readers better understand the clusters. Label the clusters to indicate their functions, such as identifying those related to ribosomal activity (see the attached figures for reference). 2) The resolution of Figures 4, 5, 6, and 8 should be improved for better readability, as some text (e.g., in Figure 6) is difficult to read. 3) Ensure consistent font styles across all figures and maintain proper text-to-figure ratios. For instance, the axis text in Figure 3 appears distorted. 4) Make the “Figure 6” in line 962 bold to match the format of other figures’ legends.

6.        The manuscript contains repeated and inconsistent explanations of abbreviations. For example, "PPI" is explained twice, on line 72 and again on line 134. "PTM" is first used on line 78 but is not explained until later (line 181) in the manuscript. Abbreviations should be defined at their first appearance and consistently used thereafter to avoid redundancy and ensure clarity. Please review the entire manuscript to ensure that all abbreviations follow this standard practice.

7.        There are numerous formatting issues, including inconsistent indentation, improper equation formatting (e.g., line 304, 1/ λ-1 should be exponent, and line 681 has a messy equation format), and the use of "we" instead of "I," given that there is only one author.

Comments on the Quality of English Language

The manuscript contains grammatical errors and unnatural phrasing. Also, there are several non-professional terms used throughout the manuscript, which should be replaced with more precise and formal language. Here are some specific examples:

  • Lines 16-18: "carry multiple attacks on single human proteins out" would be clearer as "carry out multiple attacks on individual human proteins."
  •  Lines 18-19: The sentence "Through Cytoscape we showed functional implications and its tropism to human organs/tissues, such as nervous system, liver, blood, and lungs" contains grammatical errors and could be rephrased for clarity.
  • Lines 55-56: The phrase "Therefore, because we don't know the molecular processes, we test many drugs based on symptoms" sounds awkward and should be revised for clarity.
  • Lines 69-71: The sentence "Even if we do not yet know who the vaccine interacts with, we can know who it attacks based on the molecular interactions studied experimentally" should be changed to "which it interacts with," as "who" is not appropriate in this context.
  • Lines 366-367: The phrase "The mature S1 subunit gets 711 aa, as decoded by its mRNA" should be revised to "The mature S1 subunit contains 711 amino acids," as "gets" does not sound professional.
  • I feel that the term "one-to-one interaction" is not entirely accurate in structural biology, as it usually refers to interactions between monomeric units. In this study, it may involve interactions between an S1 unit and a protein oligomer, so this should be clarified.

These examples highlight broader issues in the manuscript. A thorough proofreading is recommended to improve the overall language quality.

Author Response

To Reviewer No 1

Thank you for your patience in pointing out my linguistic discrepancies. I usually check the grammar but in a long text something slips by.

Length: The manuscript is excessively long. I feel it could benefit from streamlining to improve clarity and conciseness. While the analysis is thorough and comprehensive, some technical sections can be condensed or moved to supplementary materials to make the manuscript focus on the biological implications.

3) Streamline the discussion of network centralities and topological parameters or move these details to supplementary materials.

See joint response to reviewer No2, at the end.

Abstract: The abstract would benefit from a brief introductory sentence to provide context on the SARS-CoV-2 viruses and S1 subunit, allowing readers to better understand the context.

I added a sentence in the abstract to introduce SARS2
Introduction: 1) The introduction should be more concise. For example, the first two paragraphs can be combined to streamline the background. Also, the paragraphs explaining why S1 is being studied, the use of PPI for studying S1, and the impact of post-translational modifications on the interatomic analysis (lines 57-132) should be shortened for clarity. The effect of post-translational modifications is discussed later in the manuscript, this part can be removed from introduction to avoid repetition.

The introduction is now more concise, after the reduction of the first paragraphs. I also added some explanations about ACE2. I removed the topics on lines 57-132 and those on PTMs to avoid repetitions.

Results: 1) Consider moving Figure 1S, which shows the structure of S1 unit, to the main text (section 3.1) to help readers better understand and visualize the S1 subunit. Also, the figure would benefit from a more professional appearance. Instead of a screenshot from PyMOL, create the diagram with a white background and clearly labeled bars (see the attached figures for more information).
2) Line 375 references "pdbj"; it should be corrected to "PDB" to avoid confusion.

4) In lines 1337-1351, some terms are unnecessarily placed in quotation marks; this should be clarified or revised.

1) I inserted a figure of the complex (figure 1) with the suggested changes. I have not removed the S1 structure from the supplements because it is the only figure from which we can appreciate the notable presence of disordered segments that gives the protein its properties in solution. This is an important aspect never highlighted with clarity by others.

2) Now pdbj is PDB, anyway pdbj means PDB Japan.

4) I removed the question marks from the line 1337 to 351 (now XXXX).

  1. Figure Presentation: The presentation of figures needs improvement.

1) In the protein interactome maps, the nodes should be analyzed and categorized to help readers better understand the clusters.

All the 5 attached excel files report, according to the topic discussed and in a schematic way, a broad, and detailed functional characterization of each process, term, or node discussed in this article.

Label the clusters to indicate their functions, such as identifying those related to ribosomal activity (see the attached figures for reference).

I have reported in the Supplements (file 5S), a functional schematization of the sub-graphs of the interactome-814 which is the functionally richest.

2) The resolution of Figures 4, 5, 6, and 8 should be improved for better readability, as some text (e.g., in Figure 6) is difficult to read.

I implemented the resolution in figures 4, 5, 6, and 8 (now renumbered as 5, 6, 7, and 9) and, whenever possible, in the others.

3) Ensure consistent font styles across all figures and maintain proper text-to-figure ratios. For instance, the axis text in Figure 3 appears distorted.

I have adjusted the text in the figures using the same fonts wherever possible. I am intrigued by your statement about text axis distorted in figure 3 (now 4). I used a ruler, but it looks OK.

4) Make the “Figure 6” in line 962 bold to match the format of other figures’ legends.

Done

  1. The manuscript contains repeated and inconsistent explanations of abbreviations. For example, "PPI" is explained twice, on line 72 and again on line 134. "PTM" is first used on line 78 but is not explained until later (line 181) in the manuscript. Abbreviations should be defined at their first appearance and consistently used thereafter to avoid redundancy and ensure clarity. Please review the entire manuscript to ensure that all abbreviations follow this standard practice. FATTO

I have implemented all the tips.

  1. There are numerous formatting issues, including inconsistent indentation, improper equation formatting (e.g., line 304, 1/ λ-1 should be exponent, and line 681 has a messy equation format ????), and the use of "we" instead of "I," FATTO given that there is only one author.

I have used the first person where necessary. The form used previously (we) is because of an ancient Latin form (plural maiestatis) widely used in my language. Using the first-person plural (rather than singular) of the verb and of the corresponding personal and possessive adjectives and pronouns is very common in particular situations: in official or institutional contexts, out of courtesy, in commercial documents, in narrative prose, in professional writings when involvement is required.

The irregular indentation is because of Word. However, the final layout by the author/editor during manuscript editing will correct the irregular indentation. At that stage, the editor asks the author to make the specific adjustments.

  • Lines 16-18: "carry multiple attacks on single human proteins out" would be clearer as "carry out multiple attacks on individual human proteins."

I corrected it

  • Lines 18-19: The sentence "Through Cytoscape we showed functional implications and its tropism to human organs/tissues, such as nervous system, liver, blood, and lungs" contains grammatical errors and could be rephrased for clarity.

I rephrased the text.

  • Lines 55-56: The phrase "Therefore, because we don't know the molecular processes, we test many drugs based on symptoms" sounds awkward and should be revised for clarity.

This sentence is no longer there because the text has changed.

  • Lines 69-71: The sentence "Even if we do not yet know who the vaccine interacts with, we can know who it attacks based on the molecular interactions studied experimentally" should be changed to "which it interacts with," as "who" is not appropriate in this context.

This sentence is no longer there because the text has changed.

  • Lines 366-367: The phrase "The mature S1 subunit gets 711 aa, as decoded by its mRNA" should be revised to "The mature S1 subunit contains 711 amino acids," as "gets" does not sound professional. Gets >>> shows

I corrected the verb.

  • I feel that the term "one-to-one interaction" is not entirely accurate in structural biology, as it usually refers to interactions between monomeric units. In this study, it may involve interactions between an S1 unit and a protein oligomer, so this should be clarified.

I understand the reviewer's concerns and have included a paragraph in the Supplements to clarify. I agree that this is not a trivial concern, but I see no other solution than the one used. However, the topic required extensive clarification.

To Reviewers 1 and 2.

Both of you, with different words, ask me two similar things, streamlining the manuscript and implementing the figures.

I tried to follow both of you even if sometimes you ask for conflicting things.

This work required months of calculations, analysis, and long reflections before reaching its final organization. As you can see, there is a large amount of data (in the excel sheets and in the supplements). In the manuscript, I had to present the procedures used in extensive detail, remarking all the logical steps between one and the other to justify the use of not very common approaches such as Reverse Engineering, Data Merging and Clustering. Here, the key problem is not the biological conclusion that emerges, but the significance and correctness of the methodological and technical paths used to get a logical progression that justifies and makes significant the data got.

Of course, all this made the manuscript long, but I hope you understand the conclusions include both the effects of the vaccine and the disease, which appear quite similar. You realize, of course, that associating certain diseases with the effects of a single viral protein requires robust data and coherent analyses to avoid criticism from the scientific community. In addition, this protein is the same one used to develop vaccines. I know that as reviewers you have the problem of reaching readers who are not experts in the subject, but this is a manuscript aimed at technical experts in the sector because they are the ones who have to evaluate the implications in terms of public health and I cannot summarize much with them.

I tried to implement the individual figures as best as possible. It is not a problem of resolution. The main interactomes include hundreds of genes and it is difficult to make the characters clear if you have to reduce the size of the entire interactome to fit it on a page.

Just to point out, below is containing the excel files with the type of data presented.

1) The excel files added to the article report all the elements to have complete control of the analyses performed, the data presented and their biological functions.

Excel files 1-1060 and excel 2-814: Both report:

In sheet 1, the total list of all nodes and their degree, including the indications relating to all hub nodes.

In sheet 2, the list of all existing interactions in the interactome (34986 for X and XXX for X) with both nodes and the quantitative incidence of each source on the combined score (confidence score)

In sheet 3, I show the centrality analysis in its details.

In sheet 4, i report all the GO functional terms, one by one, with their specific genes because of functional enrichment. For each term, I report the following parameters: observed gene count, background gene count, strength, false discovery rate.

Excel File3 Reverse engineering. The file reports in sheet 1 all the interactions of the interactome-1060 with viral proteins, where external experimental data supports the interactions from a biological point of view. Sheet 2 contains all the one-to-one interactions of all viral proteins. Sheet 3 contains the multiple interactions involving all viral proteins.

Excel File4 Data Merging. Sheet 1 contains the total merging of the 1060 and 814 interactomes with the highlighting of the paired and unpaired terms. I report all the genes associated with each functional term (GO), one by one, including observed gene count, background gene count, strength, and false discovery rate. Sheet 2 lists all 68,003 genes, revealing the redundancy among them.

Excel File5 Clustering. The three sheets contain all the functional clusters and all the genes involved in each of them, calculated for the uncoupled-1060, 814 and 814+1060 terms. The sheets also include the number of each cluster, its functional composition (primary, secondary, and tertiary), and the number of genes involved.

I also added a figure that summarizes, where possible, certain functions, but in the excel files there is already everything, in the smallest details and I made them available to the journal.

In conclusion, I tried to reduce the body of the results by eliminating centralities and topological parameters (according to reviewer No 1) but this distorted the sense and logic of the manuscript. Therefore, although I accept all your suggestions, I am not available to modify the core of the manuscript. If you do not agree, I will retract the manuscript and present it elsewhere.

I apologize, it is not to contradict you because I understand you, but I think this way.

Reviewer 2 Report

Comments and Suggestions for Authors

The use of interactome modeling and reverse engineering approaches contributes meaningfully to understanding SARS-CoV-2’s impact. However, there are several areas that need further clarification, methodological improvements, and additional experimental validation before the paper can be considered for publication.

1. Most of the analysis is based on computational predictions, while the lack of robust experimental validation or literature review were discussed. My suggestion is to consider adding experimental data from past papers to the discussion. 

2. Some sections of the methods, such as the use of clustering algorithms, are not well described to allow the replication of the future study. My suggestion is to provide more details about how the number of clusters was determined and how outliers were handled. 

3. The visualization of results needs to be improved to highlight your key findings. My suggestions is to reorganize some figures for clarity and impact. For example, to include more visual summaries, like the network maps of key protein interactions, may be helpful for the audience to quickly get the significance of the findings. 

Some minor issues found, including grammar and language is lack of polishing to improve the readability. My suggestion is to revise and proof-read the manuscript to ensure the technical terms are used consistently and the sentences are straightforward. 

In summary, this study presents important findings, but it requires additional clarification and better data presentation before acceptance. 

Comments on the Quality of English Language

Needs to be improved. 

Author Response

To the reviewer No2

Thank you for your patience in pointing out my linguistic discrepancies. I usually check my grammar but in a long text something slips by.

  1. Most of the analysis is based on computational predictions, while the lack of robust experimental validation or literature review were discussed. My suggestion is to consider adding experimental data from past papers to the discussion.

All interactions between S1 and human proteins are physical and experimental in vivo (BioGRID). I selected only the most significant and low-throughput ones. For details see: 2.1 BioGRID and 2.7. You can find details in the SHPID database. However, i listed in the Excel File 1, sheet 2-interactions and in the Excel File 2, sheet 2-interactions, all the interactions present in the two interactomes as analyzed by STRING. For each interaction, I report the quantitative incidence of the 7 data sources (experimental, 4 genetic, database, text mining) from which the data originates. This information was determined by an AI-powered literature search (PubMed) of at least 10,000 articles (as also reported in tables 1 and 6). All the bibliography used for each evaluation (for each calculated interactome) is available in a downloadable Excel file. I tried to explain in Appendix-A that one of the biggest limitations of current Systems Biology is the scarcity of experimental data on biological interactions. A personal estimate is that only 5-10% of biological relationships in the literature have an experimental basis (Biochemistry or Biophysics). I think that humanly speaking, there is nothing else we can do at the moment.

  1. Some sections of the methods, such as the use of clustering algorithms, are not well described to allow the replication of the future study. My suggestion is to provide more details about how the number of clusters was determined and how outliers were handled.

To determine the value of K in the K-means clustering algorithm, I used a graphical method, the elbow method. I included this information in paragraph “2.10”.

  1. The visualization of results needs to be improved to highlight your key findings. My suggestions is to reorganize some figures for clarity and impact. For example, to include more visual summaries, like the network maps of key protein interactions, may be helpful for the audience to quickly get the significance of the findings.

I have acted in this direction, also based on the suggestions of the other reviewer, by introducing figure 1 and figure 5S, which summarise some important aspects.

Some minor issues found, including grammar and language is lack of polishing to improve the readability. My suggestion is to revise and proof-read the manuscript to ensure the technical terms are used consistently and the sentences are straightforward.

I reviewed the entire manuscript, cleaning it up and paying attention to terminology.

To Reviewers 1 and 2.

Both of you, with different words, ask me two similar things, streamlining the manuscript and implementing the figures.

I tried to follow both of you even if sometimes you ask for conflicting things.

This work required months of calculations, analysis, and long reflections before reaching its final organization. As you can see, there is a large amount of data (in the excel sheets and in the supplements). In the manuscript, I had to present the procedures used in extensive detail, remarking all the logical steps between one and the other to justify the use of not very common approaches such as Reverse Engineering, Data Merging and Clustering. Here, the key problem is not the biological conclusion that emerges, but the significance and correctness of the methodological and technical paths used to get a logical progression that justifies and makes significant the data got.

Of course, all this made the manuscript long, but I hope you understand the conclusions include both the effects of the vaccine and the disease, which appear quite similar. You realize, of course, that associating certain diseases with the effects of a single viral protein requires robust data and coherent analyses to avoid criticism from the scientific community. In addition, this protein is the same one used to develop vaccines. I know that as reviewers you have the problem of reaching readers who are not experts in the subject, but this is a manuscript aimed at technical experts in the sector because they are the ones who have to evaluate the implications in terms of public health and I cannot summarize much with them.

I tried to implement the individual figures as best as possible. It is not a problem of resolution. The main interactomes include hundreds of genes and it is difficult to make the characters clear if you have to reduce the size of the entire interactome to fit it on a page.

Just to point out, below is containing the excel files with the type of data presented.

1) The excel files added to the article report all the elements to have complete control of the analyses performed, the data presented and their biological functions.

Excel files 1-1060 and excel 2-814: Both report:

In sheet 1, the total list of all nodes and their degree, including the indications relating to all hub nodes.

In sheet 2, the list of all existing interactions in the interactome (34986 for X and XXX for X) with both nodes and the quantitative incidence of each source on the combined score (confidence score)

In sheet 3, I show the centrality analysis in its details.

In sheet 4, i report all the GO functional terms, one by one, with their specific genes because of functional enrichment. For each term, I report the following parameters: observed gene count, background gene count, strength, false discovery rate.

Excel File3 Reverse engineering. The file reports in sheet 1 all the interactions of the interactome-1060 with viral proteins, where external experimental data supports the interactions from a biological point of view. Sheet 2 contains all the one-to-one interactions of all viral proteins. Sheet 3 contains the multiple interactions involving all viral proteins.

Excel File4 Data Merging. Sheet 1 contains the total merging of the 1060 and 814 interactomes with the highlighting of the paired and unpaired terms. I report all the genes associated with each functional term (GO), one by one, including observed gene count, background gene count, strength, and false discovery rate. Sheet 2 lists all 68,003 genes, revealing the redundancy among them.

Excel File5 Clustering. The three sheets contain all the functional clusters and all the genes involved in each of them, calculated for the uncoupled-1060, 814 and 814+1060 terms. The sheets also include the number of each cluster, its functional composition (primary, secondary, and tertiary), and the number of genes involved.

I also added a figure that summarizes, where possible, certain functions, but in the excel files there is already everything, in the smallest details and I made them available to the journal.

In conclusion, I tried to reduce the body of the results by eliminating centralities and topological parameters (according to reviewer No 1) but this distorted the sense and logic of the manuscript. Therefore, although I accept all your suggestions, I am not available to modify the core of the manuscript. If you do not agree, I will retract the manuscript and present it elsewhere.

I apologize, it is not to contradict you because I understand you, but I think this way.

Round 2

Reviewer 1 Report

Comments and Suggestions for Authors

Thank you for considering some of my suggestions. Adding a figure showing the structure of the SARS-CoV-2 spike protein in complex with ACE2 has greatly improved the readability and helped clarify the main points. I understand that due to the complexity of the subject, it’s not possible to simplify the main text further, but the additional structure and visual aid have certainly enhanced the flow.

Author Response

Thank you for understanding my reasons for avoiding a major reworking of the manuscript. I have carefully revised the editing of the manuscript. I think it is much more fluent now. I have added some citations suggested to me by reviewer No2.

Reviewer 2 Report

Comments and Suggestions for Authors

The authors addressed most of my comments and I would suggest them to proofread the whole manuscript before the final acceptance. Also, the background will be better if relevant references could be cited, as they are very relevant to the research, for example: 

A Comparative Analysis of SARS-CoV-2 Variants of Concern (VOC) Spike Proteins Interacting with hACE2 Enzyme

Peptide-Based Inhibitors of Protein–Protein Interactions (PPIs): A Case Study on the Interaction Between SARS-CoV-2 Spike Protein and Human Angiotensin-Converting Enzyme 2 (hACE2)

1-L Transcription of SARS-CoV-2 Spike Protein S1 Subunit

Author Response

I have added with a brief comment some very recent citations, including those suggested to me. I have revised the editing of the entire manuscript and I think it is much more fluent now, more English and less Italian. Thanks for your suggestions.